# Reactive oxygen species in endothelial signaling in COVID-19: Protective role of the novel peptide PIP-2

**Oindrila Paul[1], Isha K. Akolia[1], Jian Qin Tao[1], Nikita Jain[1], Natalia Louneva[2], Kathleen T. Montone[3], Aron B. Fisher[2], Chamith S. Rajapakse[4,5], Christian Bermudez[6], Shampa Chatterjee[1] ***

**1** Institute for Environmental Medicine and Department of Physiology, Philadelphia, Pennsylvania, United States of America, **2** Peroxitech Inc., Philadelphia, Pennsylvania, United States of America, **3** Department of Pathology, Philadelphia, Pennsylvania, United States of America, **4** Department of Radiology, Philadelphia, Pennsylvania, United States of America, **5** Department of Orthopedic Surgery, Philadelphia, Pennsylvania, United States of America, **6** Department of Surgery, University of Pennsylvania School of Medicine, Philadelphia, Pennsylvania, United States of America

* shampac@pennmedicine.upenn.edu

**Data Availability Statement:** This manuscript's minimal data set is publicly available via Figshare. The data and detailed methods have been made available in a via the following DOI or Figshare link.

## Abstract

### Introduction

Recent research suggests that endothelial activation plays a role in coronavirus disease 2019 (COVID-19) pathogenesis by promoting a pro-inflammatory state. However, the mechanism by which the endothelium is activated in COVID-19 remains unclear.

### Objective

To investigate the mechanism by which COVID-19 activates the pulmonary endothelium and drives pro-inflammatory phenotypes.

### Hypothesis

The "inflammatory load or burden" (cytokine storm) of the systemic circulation activates endothelial NADPH oxidase 2 (NOX2) which leads to the production of reactive oxygen species (ROS) by the pulmonary endothelium. Endothelial ROS subsequently activates pro-inflammatory pathways.

### Methods

The inflammatory burden of COVID-19 on the endothelial network, was recreated *in vitro*, by exposing human pulmonary microvascular endothelial cells (HPMVEC) to media supplemented with serum from COVID-19 affected individuals (sera were acquired from patients with COVID-19 infection that eventually died. Sera was isolated from blood collected at admission to the Intensive Care Unit of the Hospital of the University of Pennsylvania). Endothelial activation, inflammation and cell death were assessed in HPMVEC treated with serum either from patients with COVID-19 or from healthy individuals. Activation was monitored by measuring NOX2 activation (Rac1 translocation) and ROS production;

DOI: 10.6084/m9.figshare.25658775 Figshare link: https://figshare.com/articles/dataset/_b_Reactive_Oxygen_Species_in_Endothelial_Signaling_in_COVID-19_Protective_role_of_the_novel_peptide_PIP-2_b_/25658775.

**Funding:** S.C. was supported by NIH R56 HL139559 and R41HL164161. The funders had no role in study design, data collection and analysis, decision to publish, or preparation of the manuscript.

**Competing interests:** A.B.F. is a shareholder in Peroxitech LLC, a company that is developing PIP-2 as a therapeutic agent. O.P., S.C. and A.B.F. have filed a U.S. Provisional Patent Application titled "Protective Agent Against Endothelial Dysfunction", application No. No. 63/469,255 filed May 26, 2023. No related or modified products are applicable or filed. This patent application does not affect our adherence to PLOS ONE policies on sharing data, data acquisition parameters, and materials.

inflammation (or appearance of a pro-inflammatory phenotype) was monitored by measuring the induction of moieties such as intercellular adhesion molecule (ICAM-1), P-selectin and the NLRP3 inflammasome; cell death was measured via SYTOX™ Green assays.

## Results

Endothelial activation (i.e., NOX2 activation and subsequent ROS production) and cell death were significantly higher in the COVID-19 model than in healthy samples. When HPMVEC were pre-treated with the novel peptide PIP-2, which blocks NOX2 activation (via inhibition of $Ca^{2+}$-independent phospholipase A2, $aiPLA_2$), significant abrogation of ROS was observed. Endothelial inflammation and cell death were also significantly blunted.

## Conclusions

The endothelium is activated during COVID-19 via cytokine storm-driven NOX2-ROS activation, which causes a pro-inflammatory phenotype. The concept of endothelial NOX2-ROS production as a unifying pathophysiological axis in COVID-19 raises the possibility of using PIP-2 to maintain vascular health.

## Introduction

An early event in coronavirus disease 2019 (COVID-19) is the entry of the severe acute respiratory syndrome coronavirus 2 (SARS-CoV-2) virus into pulmonary epithelial cells via surface angiotensin-converting enzyme 2 (ACE2) receptors, resulting in viral infection, which, in turn, activates a systemic inflammatory response [1–3]. While epithelial infection and injury are well-characterized events in SARS-CoV-2 attack (and indeed with all viral attacks), there is increasing evidence of endothelial dysfunction as well. Indeed, elevated plasma levels of endothelial cell activation biomarkers have been reported to predict poor outcomes [4–8]. Specifically, two proteins released by activated endothelial cells (EC), namely the von Willebrand Factor (vWF) antigen and soluble P-selectin, are significantly elevated in patients with COVID-19 [8, 9].

We previously reported that EC activation occurs via redox signals such as reactive oxygen species (ROS) emanating from the EC membrane [10–12]. We and others have also reported that ROS may play a role in the induction of the endothelial pro-inflammatory phenotype [13, 14]. In a post-mortem analysis of lung tissue from individuals who died of COVID-19, we noted a pro-inflammatory phenotype in the EC layer [15]. There have also been reports of endothelial activation, inflammation, and injury in COVID-19 [5–8]. These observations are supportive of the fact that SARS-CoV-2 induces EC activation and drives a pro-inflammatory phenotype that presumably affects clinical outcomes. However, the mechanism by which ECs are activated or inflamed (i.e., there is an appearance of a pro-inflammatory phenotype in ECs) with COVID-19 is not very clear.

Previously we reported that human pulmonary microvascular endothelial cells (HPMVECs) and ECs of other vascular beds respond to various inflammatory stimuli by activation of the NADPH oxidase 2 (NOX2) pathway, which leads to ROS production [16]. Thus, we hypothesized that endothelial activation during COVID-19 occurs via NOX2 activation, and that the resultant ROS produced induces a pro-inflammatory phenotype in EC. To test this hypothesis, NOX2 activation and the regulation of a pro-inflammatory phenotype in EC

were studied using an *in vitro* model of HPMVEC that recreates the "inflammatory load or burden" of the systemic circulation of COVID-19 patients.

The concept of endothelial NOX2-ROS production as a unifying pathophysiological axis that drives EC inflammation (and injury) during COVID-19 suggests the possibility of designing therapeutics to target ROS via NOX2 blockade. Our previous investigations into the activation of the NOX2 pathway had revealed that the calcium-independent phospholipase $A_2$ activity (aiPLA$_2$) of peroxiredoxin 6 (Prdx6) was pivotal for NOX2 activation (and ROS generation) of lung endothelial cells [17, 18]. Furthermore, we discovered that the aiPLA$_2$ activity of Prdx6 occurred via a catalytic site (S32-H26-D140 triad) [19, 20]. Based on this observation, we developed a 9-aminoacid oligopeptide peroxiredoxin 6-inhibitory peptide-2 (PIP-2), which binds to the catalytic triad of Prdx6 and prevents NOX2 assembly [21–23]. Here we evaluated: 1) the effect of COVID-19 on EC activation (NOX2 activation and ROS production), 2) the role of NOX2-ROS in driving the pro-inflammatory phenotype of EC and 3) the effect of PIP-2 on these signaling events.

## Materials and methods

### Post-mortem tissue

Lung samples from five patients who died of COVID-19 acute respiratory distress syndrome (between April and Oct 2020) and five patients who died of non-COVID complications were analyzed. Non-COVID samples were obtained from patients who died prior to 2019 (Oct 2018 to Dec 2018) from non-respiratory causes (cardiac transplantation, breast cancers, renal cancer, and sarcoid disease diagnosed at autopsy). Written informed consent was obtained from the next of kin of the deceased by the Department of Pathology, University of Pennsylvania School of Medicine, for postmortem examination. For patients with COVID-19, SARS-CoV-2 infection was confirmed by real time PCR analysis using the Cepheid Gene Xpert RT PCR assay (Cepheid, Sunnyvale, CA 94089). Autopsies were performed by trained personnel using personal protective equipment in accordance with the recommendations of the University of Pennsylvania School of Medicine. All samples were de-identified and supplied to the authors in November 2020. The authors were not provided any access to identifiable participant information.

### Blood and serum samples

Blood was collected by the Institute for Immunology at the University of Pennsylvania as part of the Molecular Epidemiology of SepsiS in the ICU (MESSI-COVID) study. The samples selected for this study were from 4–5 patients with COVID-19 who were admitted to the ICU and eventually died. Blood (Sera) were collected between April to July 2020 from patients at admission. Written informed consent was obtained for collecting samples that were collected after approval by the Institutional Review Board of the University of Pennsylvania. All samples were de-identified.

### Measurement of serum TNF-α

TNF-α concentrations were measured in sera collected from healthy controls and patients with COVID-19. The human TNF- α uncoated ELISA kit (Invitrogen, Bender Med Systems GmBH, Vienna, Austria) was used to measure TNF-α levels within a detection range of 0.0128–5000 pg/mL. Standard curves were generated by making serial dilutions of a known concentration of TNF-α. The absorbance of sera from patients with COVID-19 and healthy controls were measured using an ELISA microplate reader (Infinite F50 Plus, Tecan,

Männedorf Switzerland). The absolute concentration of TNF- α was calibrated using standard curves as suggested by the manufacturer.

## Immunostaining lung sections

Lung sections were dual labeled to measure the expression on endothelial layer for platelet endothelial cell adhesion molecule (PECAM-1) and intercellular adhesion molecule (ICAM-1). PECAM-1 (labeled with red fluorescence) served as an EC marker, whereas ICAM-1 (labeled with green fluorescence) served as an index of EC inflammation or pro-inflammatory phenotype of EC. Colocalization of these two signals was used to identify the ICAM-1 expression of the EC layer. Lung sections were deparaffinized and after antibody retrieval, these sections were stained using a monoclonal antibody to ICAM-1 and a polyclonal anti-PECAM-1-conjugated to Alexa Fluor 594. For ICAM-1 detection, the secondary antibody used was conjugated to Alexa Fluor 488 at a dilution of 1:200 (Life Technologies, Eugene, OR, USA). For each sample, 4 sections were stained and imaged using a confocal microscope. Within each section, five fields were imaged and analyzed. Fluorescence signals were quantified as follows: first, the areas with red fluorescence signals were outlined. Next, the intensity of the green fluorescence within that area was measured by Image J software. Finally, the green intensity was normalized to the area (that showed red fluorescence). The normalized values (expressed as arbitrary fluorescence units) were considered to represent the ICAM-1 expression in the endothelium.

## *In vitro* COVID-19 model

Immortalized human pulmonary microvascular endothelial cells (HPMVEC clone ST1.6R) were a gift to one of the authors (S.C) from Prof. Kirkpatrick as previously described [24]. HPMVEC were grown in T-25 culture flasks and routinely maintained in Dulbecco's Modified Eagle's Medium (DMEM) supplemented with 10% fetal bovine serum (FBS), 1x glutamax, 1x antibiotics, and endothelial cell growth supplement (Millipore Upstate). The cells were grown and maintained in 5% $CO_2$ at 37˚C. Cells were plated at a density of 30–35 cells/mm$^2$ either on Greiner Bio-One Cell View dishes (with a glass bottom to enable fluorescence imaging of ROS-dye or SYTOX™-Green) or on coverslips (Fisherbrand™ Pittsburgh, PA, USA) that enable easy cell fixation. After seeding and attachment, HPMVEC were pre-treated with PIP-2 (encapsulated in liposomes; blank liposomes were used as controls) after which they were either exposed to media supplemented with serum from patients with COVID-19 (referred to as COVID-19 serum) or serum from healthy individuals (referred to as normal serum). Treatment with COVID-19 serum led to the apoptosis of HPMVEC; therefore, at the completion of each of these incubation steps (Serum, PIP-2 etc.), apoptotic cells were removed by lightly washing the coverslips with medium (DMEM + FBS) to remove dead and non-adherent cells. This is a crucial step in immunostaining and ROS imaging experiments. The cell preparation was monitored for endothelial activation (ROS production and Rac1 translocation), endothelial inflammation (ICAM-1, NLRP3, P-selectin), and cell death.

## ICAM-1 and NLRP3 assays

HPMVEC were incubated for either 1 h (for ROS, Rac translocation) or 2 h (for ICAM, NLRP3, caspase-1) or 30 min (for SYTOX™ Green cell death assay) in the incubator at 37˚C and 5% $CO_2$ with medium supplemented with 10% COVID-19 serum or normal serum. Different incubation times were used for ROS and for the ICAM/NLRP3 assays as ROS production occurs rapidly upon stimulation of ECs, whereas expression of ICAM-1 and NLRP3

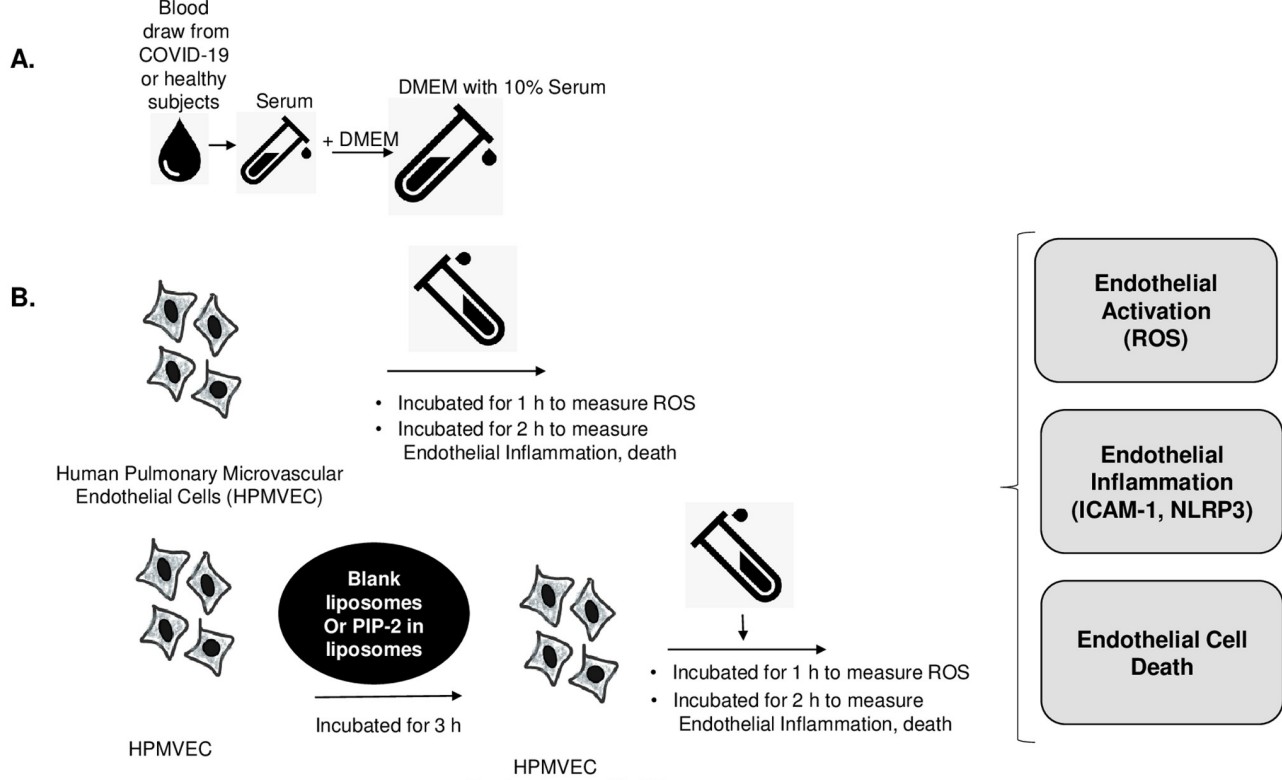

**Fig 1. Experimental design of this study.** Human pulmonary microvascular endothelial cells (HPMVECs) were exposed to sera obtained from patients with COVID-19. Samples were from those patients who were admitted to the ICU and eventually died. A. Sera from patients with COVID-19 (COVID-19 serum) or healthy individuals (normal serum) were used to supplement Dulbecco's modified Eagle's Medium (DMEM). B. The in vitro HPMVEC COVID-19 model: HPMVECs were exposed to media supplemented with COVID-19 serum or normal serum. In separate experiments, HPMVECs were pre-treated with either NADPH oxidase 2 (NOX2) inhibitor PIP-2 encapsulated in liposomes (PIP-2 carrying liposomes) or blank liposomes. This was followed by exposure to COVID-19 serum or normal serum. All cells were monitored for endothelial activation (i.e., ROS production), endothelial inflammation (i.e., appearance of pro-inflammatory phenotype), and cell death.

subunit requires longer periods as reported previously [13]. A shorter incubation period (30 min) was used to monitor cell death. The protocol is illustrated in Fig 1.

## ROS production assays

HPMVEC were treated with a medium supplemented with COVID-19 serum or normal serum for 1 h. The medium was then replaced with fresh DMEM without FBS, and the cells were labeled with the ROS sensitive dyes CellROX™ green (ThermoFisher, Carlsbad, CA, USA) or dihydroethidium (DHE) (Sigma Aldrich, Saint Louis, MO, USA). **For CellROX assay**: HPMVEC were incubated with CellROX™ Green (10 μM) for 30 min at 37°C and 5% $CO_2$. Cells were imaged for fluorescence of the oxidized dye ($\lambda_{ex}$ 488 nm/$\lambda_{em}$ 520 nm). **For DHE assay**: HPMVEC were incubated with DHE (10 μM) for 20 min. The cells were imaged for 2-hydroxy-ethidine fluorescence using a confocal microscope ($\lambda_{ex}$ 520 nm/$\lambda_{em}$ 590–620 nm). For each dye, all images were acquired using the same exposure and acquisition settings, as previously reported [10, 25].

## PIP-2 synthesis

PIP-2 was synthesized by APeptide Corp. (Shanghai, China) as a trifluoroacetate salt with >90% purity as shown by both mass spectrometry and HPLC results provided by the

manufacturer. The amino acid sequence of PIP-2, LHDFRHQIL, has been published previously [21–23]. The peptide was stored at -20˚C although it has been shown to have long term stability (>10 months) for storage at room temperature. For *in vitro* administration into HPMVEC, the peptide was encapsulated in liposomes [dipalmitoyl phosphatidylcholine (DPPC): egg PC: egg phosphatidylglycerol (PG): cholesterol (0.5:0.25:0.10:0.15, molar fraction)] as previously described [26]. The liposomes with encapsulated PIP-2 were suspended in 14 mg lipid and 2 mg PIP-2 per ml of phosphate-buffered saline (PBS); liposomes were stored overnight at 4˚C and used within 24 h of preparation. The final concentration of PIP-2 in the medium was 40 μg/ml. Blank liposomes were used as control and liposomes containing PIP-2 (referred to as PIP-2 carrying liposomes) were prepared, stored and used simultaneously.

## PIP-2 pre-treatment

HPMVEC were pre-treated for 3 h at 37˚C with medium supplemented with either blank liposomes or liposomes containing PIP-2. Next, the medium was replaced with fresh medium supplemented with COVID-19 or healthy controls for 1 h (for ROS measurement) or 2 h (for measuring ICAM-1, NLRP3 and P-selectin protein expression). This fresh medium also contained either blank liposomes or PIP-2 carrying liposomes, such that liposomes were available in the medium throughout the experiment.

## Rac translocation assessment

The localization of Rac1 GTPase on the EC membrane was monitored in COVID-19 serum or normal serum-treated HPMVEC by using monoclonal anti-Rac-Alexa488 at 1:100 (EMD Millipore, Dermstadt, Germany). The membranes were visualized using a polyclonal anti-flotillin antibody (1:200; Proteintech, Rosemont, IL, USA). Rac translocation was assessed based on the fluorescence intensity of Rac1 (green) on the EC membrane. This was performed by measuring the intensity of the fluorescent signal at the endothelial cell-cell border using ImageJ. For each field, regions of interest (ROI) with three cells were selected. For each ROI, the cell perimeters were outlined (as dotted lines). The fluorescence intensity of the green (Rac) signal along the dotted line was obtained and then normalized to the cell perimeter.

## Immunostaining of ICAM-1 and P-selectin

HPMVEC were treated with COVID-19 serum or normal serum (as described in the section *in vitro COVID-19 model*) for 2 h were washed with 1x PBS and were then fixed with 4% paraformaldehyde for 10 minutes followed by permeabilization. Permeabilized cells were immunostained with anti-ICAM at 1:150 (Abcam, Waltham, MA, USA) or anti-P-selectin at 1:100 (SantaCruz Biotechnology, Dallas, TX, USA) which reacts with the human proteins. The secondary antibody was goat-anti mouse IgG conjugated to Alexa 488 (green).

## Immunostaining of NLRP3 inflammasome

NLRP3 subunit and caspase-1 were immunostained using monoclonal anti-NLRP3 (R&D Systems, Minneapolis, MN, USA) and anti-caspase-1 (for activated caspase-1; Novus Biologicals, Centennial, CO, USA) at a 1:500 dilution.

## SYTOX™ Green cell death assay

Cell viability assays were performed using the SYTOX™ Green Nucleic Acid Stain (Thermo Fisher Scientific). Cells were treated COVID-19 or normal serum and then labeled with 100

nM SYTOX Green. Culture dishes were maintained under temperature-controlled conditions on the stage of a confocal microscope and imaged at 1 and 30 min.

### Confocal microscopy

For all the immunostaining experiments, appropriate nonimmune IgG controls were used to set the exposure settings for image acquisition. VECTASHIELD antifade mounting medium containing DAPI (nuclear stain) was obtained from Vector Laboratories (Burlingame, CA, USA). Fluorescence imaging was performed either by epifluorescence microscopy using a Nikon TMD epifluorescence microscope, equipped with a Hamamatsu ORCA-100 digital camera, and MetaMorph imaging software (Universal Imaging, West Chester, PA, USA) or by using a Leica TCS SP8 confocal microscope with super-resolution imaging of green and red dyes (with adjusted laser power and sequential scanning with separate detection channels to avoid bleed through). Fluorescence images were acquired at $\lambda_{ex}$ = 488 nm for green and $\lambda_{ex}$ = 520 nm for red; all images were acquired with the same exposure and acquisition settings as reported previously [15, 25]. LASX and MetaMorph software were used for image acquisition.

### Image data analysis

Quantification of the fluorescence signal was performed using the Image J or MetaMorph software. All analyses were performed in a single-blind manner. The integrated Intensities were normalized either to the field area (as we previously reported [10, 25]), the signal in ECs or the EC perimeter. A characteristic feature of the quantification of cellular fluorescence is that an image in which the cellular signal intensities range from weak to strong within a field, is quantitatively comparable to that in which all cells show average intensity throughout the field. These variations were captured as the data were quantified over several fields within a sample or section.

### Statistical analysis

Results are presented as mean ± standard deviation (SD) of n = 5. Several (3–4) independent experiments were performed for each serum sample. Group differences were evaluated using a 2-tailed t-test or one-way ANOVA followed by a post hoc *t*-test as appropriate. Statistical significance for all studies was set as $p < 0.05$.

### Ethical approval and consent

Written informed consent was obtained from the next of kin for postmortem examination.

## Results

### Increased ICAM-1 expression along the vascular wall in the lungs of patients with COVID-19

A pivotal moiety that participates in the lung inflammatory response is ICAM-1. Thus, we monitored ICAM-1 in the vascular walls of human lung sections from patients who died of COVID-19. Compared with the lungs of patients with non-pulmonary diseases, ICAM-1 levels in patients with COVID-19 were significantly higher. Fig 2A shows a representative fluorescence micrograph of ICAM-1 expression (green) along the pulmonary vessel wall of COVID-19 and non-COVID-19 patients. Owing to the extensive vascular damage in the lungs, identification of intact capillaries and arteries was not possible. Therefore, PECAM-1 which is widely recognized as an EC marker used to identify the endothelial layer. All structures that showed red fluorescence (PECAM-1 positive) were identified as endothelium. Both ICAM-1 (green)

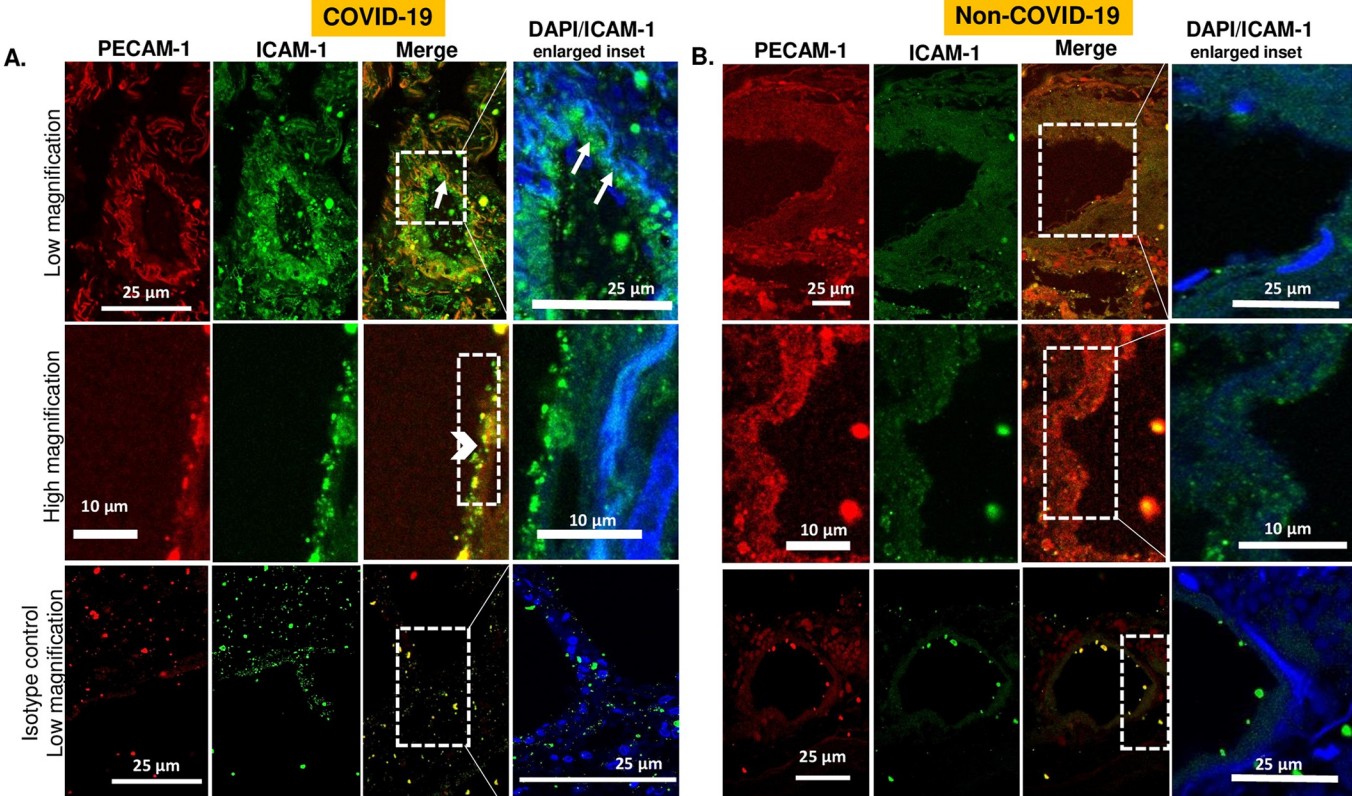

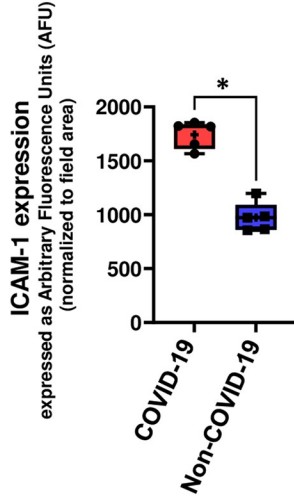

**Fig 2. ICAM-1 expression in postmortem lung tissue of COVID-19 affected and non-COVID-19 affected individuals.** Representative images of the immunofluorescence in lung sections are displayed in four vertical panels. Sections were stained with ICAM-1 (green, first vertical panel), PECAM-1 (red, endothelial marker, second vertical panel). The third vertical panel is a merged panel (yellow) which shows the extent of ICAM-1 expression in the endothelium. In the fourth vertical panel, insets are magnified to show ICAM-1 expression (green) around individual cells (DAPI nuclear stain, blue). A. Lung section from a COVID-19 affected individual. Upper horizontal panel: Representative field showing a blood vessel at low magnification. Marged panel shows ICAM-1 colocalized along the endothelial layer (yellow). Magnified inset shows ICAM-1 expression (green) around the cells of the vessel wall (white arrow). Middle horizontal panel: Higher magnification images along a small section of a vessel wall. Yellow of the merged panel indicates ICAM-1 on the endothelial layer (white arrowhead). Lower horizontal panel: isotype control (non-immune antibody of the same type/dilution as the anti-ICAM-1 or anti-PECAM-1 used in the upper and middle panels). B. Lung section from an individual without respiratory disease. Upper and Middle horizontal panels: Lower and higher magnification

images along a small section of a vessel wall show low ICAM-1 expression (green). Lower horizontal panel: isotype control (non-immune antibody of the same type/dilution as the anti-ICAM-1 or anti-PECAM-1 used in the upper-middle panels). C. Quantification of ICAM-1 expression using MetaMorph Imaging Software. Data are from postmortem lung tissue of five COVID-19-affected individuals and five non-COVID-19-affected (cardiac transplant, breast cancers, renal cancer, sarcoid diagnosed at autopsy) individuals. Four sections were stained and imaged for each sample. Within each lung section, five fields were imaged and analyzed, and the fluorescence intensity of the green signal (representing ICAM-1) within the red region was quantified. For this, the red fluorescent areas were first outlined. Next, the green fluorescence intensity within that area was measured. This integrated intensity was normalized to the area (that had red fluorescence). This was considered ICAM-1 around the endothelium. Data are shown as a box & whiskers plot. The group average and median are indicated by a plus sign and horizontal bar, respectively. Data are obtained from n = 5 each of COVID-19-affected and non-COVID-19-affected individuals. *p<0.01 as compared to non-COVID affected lungs. Group differences were evaluated using ANOVA followed by a post hoc t-test.

and PECAM-1 (red) expression as detected by yellow color was observed (Fig 2A, upper horizontal panel) in the COVID-19 positive samples. Better visualization of the localization of ICAM-1 (green fluorescence) along the endothelium (red fluorescence of PECAM-1) was attained when images were acquired at higher magnification (middle horizontal panel) where autofluorescence characteristics (emanating from fibers of the connective tissues of blood vessels) were minimized. ICAM-1 expression in non-COVID lungs was lower compared to that in COVID-19 lungs (Fig 2B). ICAM-1 expression was monitored based on the integrated intensity of the green fluorescence normalized to the area covered by the red fluorescence. Fig 2C shows that ICAM-1 expression was significantly higher in the lungs of patients with COVID-19 than in non-COVID affected individuals (non-respiratory cases).

## Endothelial inflammation as assessed by ICAM-1, NLRP3, and P-selectin in the *in vitro* HPMVEC COVID-19 model

The pro-inflammatory phenotype of ECs involves the expression of certain inflammatory moieties that can promote the adherence of immune cells in the circulation to the endothelial layer. Here we assessed the endothelial inflammation (i.e., the pro-inflammatory phenotype of ECs) by monitoring ICAM-1, NLRP3 and P-selectin in HPMVECs in an *in vitro* COVID-19 model. HPMVECs treated with COVID-19 serum were found to have a pro-inflammatory phenotype compared to cells treated with normal serum.

HPMVEC exposed to normal serum showed very minimal ICAM-1 and NLRP3 subunit expression (green fluorescence arising from secondary antibody) as compared to cells treated with COVID-19 serum (Fig 3A and 3C). ICAM-1 and NLRP3 expressions were quantified from the fluorescence intensity. Using the Image J, integrated intensities were normalized to the area of cells within the field as we have previously reported [25, 27]. Fig 3B and 3D show the quantification of ICAM-1 and NLRP3 expressions in HPMVEC. A significant difference in both moieties between the normal serum and COVID-19 serum treated models was observed. P-selectin, another EC adhesion molecule, that mediates the binding of neutrophils and monocytes, had elevated levels in the COVID-19 model, compared to normal serum (healthy) controls, but the difference was not statistically significant (Fig 3E).

## Endothelial cell death in the *in vitro* HPMVEC COVID-19 model

COVID-19 is increasingly considered an endothelial disease, as EC death is a characteristic feature of severe or fatal COVID-19. Therefore, we assessed EC death in the *in vitro* cellular model of COVID-19. We observed significantly greater cell death in ECs treated with COVID-19 serum than in those treated with normal serum. Cell death was monitored by labeling HPMVEC with SYTOX™ Green, which does not cross intact membranes but penetrates compromised membranes and binds to nucleic acids [28] within the cell resulting in fluorescence

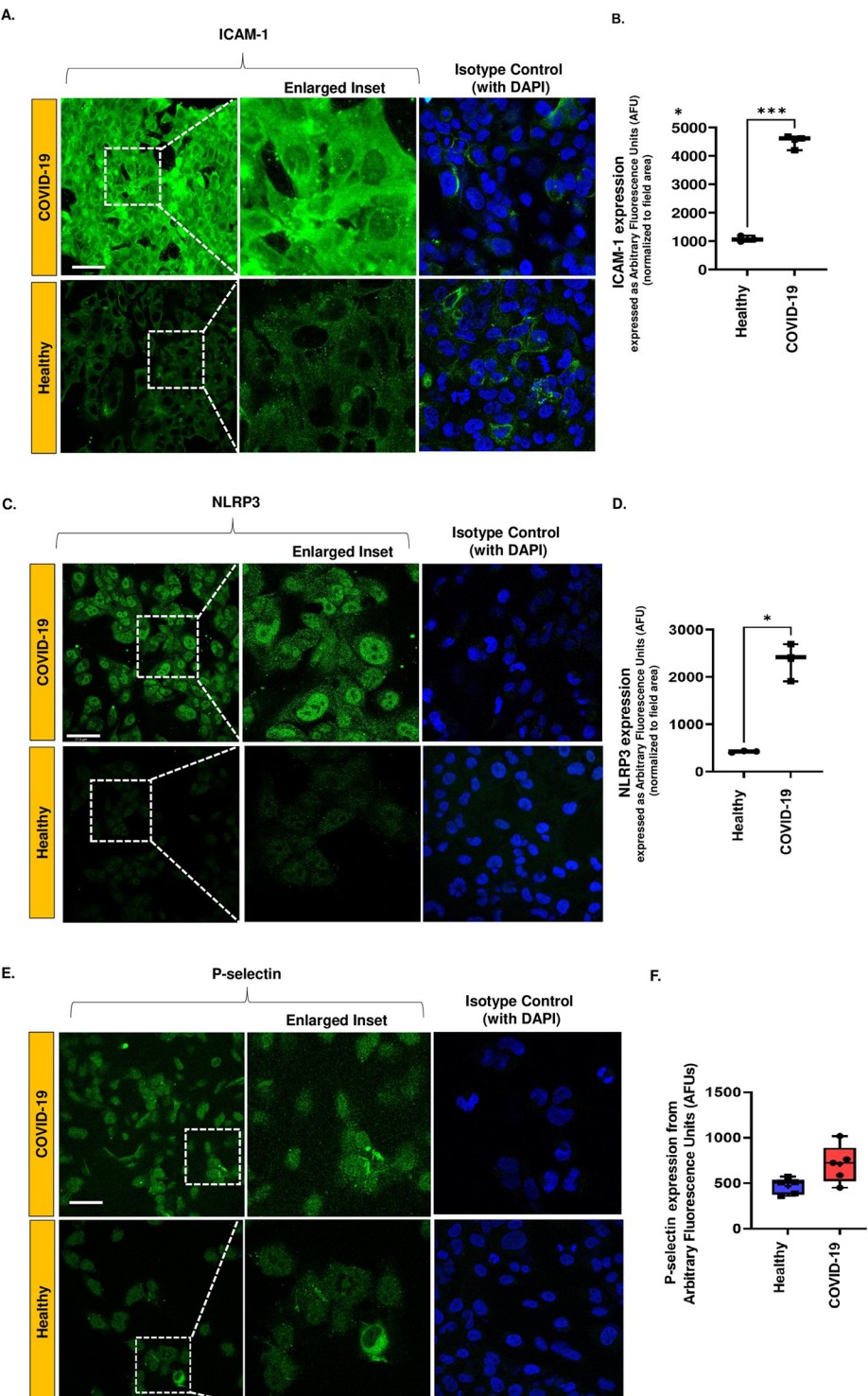

**Fig 3. Expression of inflammatory moieties in an in vitro HPMVEC COVID-19 model.** For all immunostaining experiments, HPMVEC were exposed to COVID-19 or normal serum. This implies media supplemented with COVID-19 serum or normal serum as described in Materials and Methods Section. HPMVEC were washed after serum exposure, fixed, and immunostained. Images were acquired using confocal microscopy. Scale bar is 25 μm. A. ICAM-1 expression in representative images of HPMVECs after a 2 h exposure to COVID-19 serum or normal serum. Isotype controls are included. Due to low fluorescent signals in the isotype staining, DAPI labeling is provided to visualize cells in the field. B. ICAM-1 expression quantification by Image J. The fluorescence intensity was normalized

to the field area. C. NLRP3 subunit in representative images of HPMVECs after a 2 h exposure to COVID-19 serum or normal serum. Isotype controls are included. Due to low fluorescent signals in the isotype staining, DAPI labeling is provided to visualize cells in the field. D. NLRP3 expression quantification from intensity of fluorescent signal by Image J. The fluorescence intensity was normalized to the field area. Inset is enlarged for both A. and C. to show expression of ICAM-1 and NLRP3 in the cytoplasm/perinuclear and nuclear membranes respectively. E. P-selection expression in representative images of HPMVECs after a 2 h exposure to COVID-19 serum or normal serum. Isotype controls are included. Due to low fluorescent signals in the isotype staining, DAPI labeling is provided to visualize cells in the field. F. P-selection expression quantification by Image J. The fluorescence intensity was normalized to the field area. Data in B, D, and F. show the integrated fluorescence intensity obtained as integrated intensity in a field normalized to the field area (i.e., sum of the area occupied by cells within the field). Data are shown as a box & whiskers plot. The group average and median are indicated by a plus sign and horizontal bar, respectively. Data for these experiments were obtained from N = 4–5 individuals in each category (COVID-affected or healthy individuals). Three separate experiments were conducted for each individual's serum sample and the average of the three trials was considered to represent the values for that sample. *p≤ 0.01 when compared to healthy (normal) serum.

(Fig 4A). The fluorescence signal showed that ECs in the COVID-19 model, had significantly higher cell death rates than control (Fig 4B).

## Endothelial ROS production in the *in vitro* HPMVEC COVID-19 model

To identify the signals that may drive the endothelial inflammation in the HPMVEC model of COVID-19, we evaluated amount of ROS production. 1-hour after exposure to (media supplemented with) COVID-19 serum or normal serum, HPMVEC were labeled with ROS sensitive dyes, CellROX™ Green or Dihydroethidium (DHE). CellROX™ green is a cell-permeable non-fluorescent compound in its reduced form that is oxidized by ROS ($H_2O_2$) to emit green

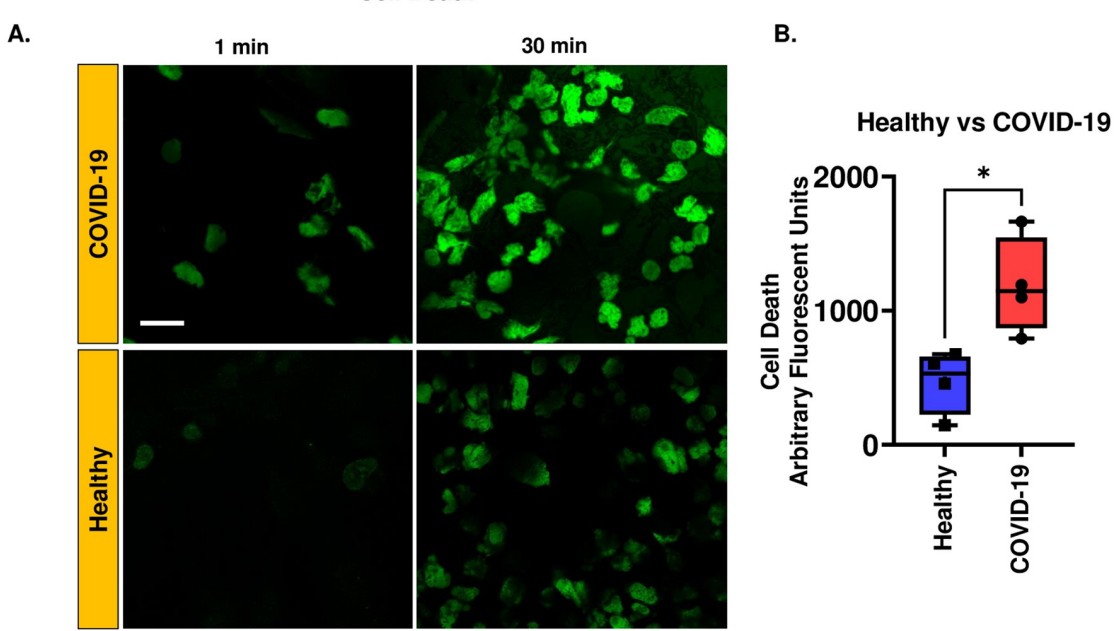

**Fig 4. Endothelial cell death in the in vitro HPMVEC COVID-19 model.** A. HPMVEC were labeled with SYTOX™ Green (100 nM) and exposed to COVID-19 serum or normal serum. This implies media supplemented with COVID-19 serum or normal serum as described in Materials and Methods Section. Representative images of the same field were acquired at 1 and 30 min by confocal microscopy. Scale bar = 25 μm. B. Quantification of the fluorescent signal from SYTOX™ Green was carried out by Image J. The integrated intensity of all cells in the field was normalized to the field area (area of the cells). Data are shown as a box & whiskers plot. Data were obtained from N = 4 individuals in each category (COVID-19 affected and healthy individuals). *p<0.01 as compared to healthy serum.

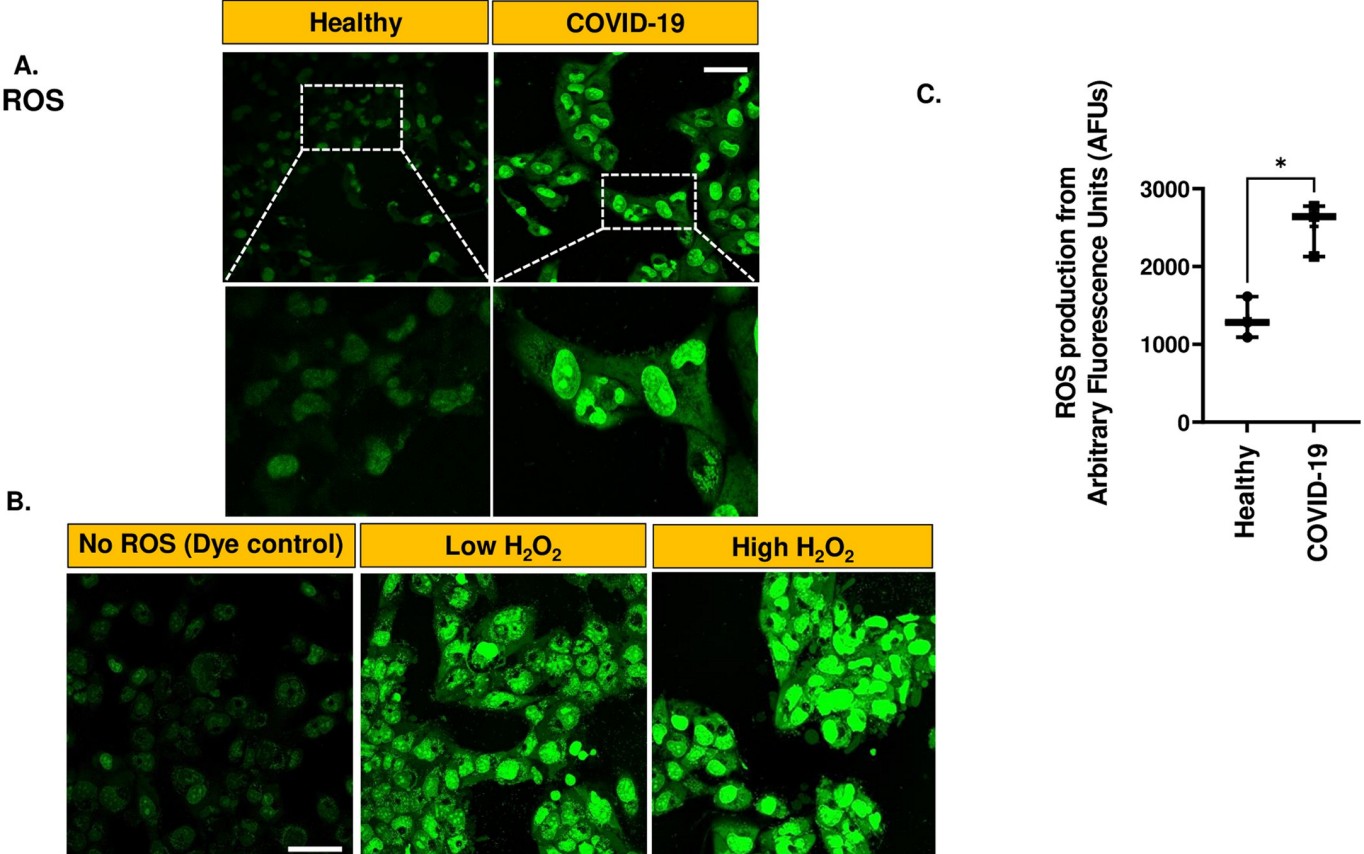

**Fig 5. ROS production in the in vitro HPMVEC COVID-19 model.** A. HPMVEC were exposed for 1 h COVID-19 serum and normal serum. This implies media supplemented with COVID-19 serum or normal serum as described in Materials and Methods Section. HPMVEC were labeled with CellROX™ Green (10 μM) for 30 min and imaged for fluorescence at λex 488 nm. Scale bar is 25 μm. B. Negative and Positive Controls to observe (CellROX)ROS fluorescence patterns in ECs in response to high and low ROS: Untreated and $H_2O_2$ treated (final concentration of $H_2O_2$ for low = 50 μM; high = 500 μM). After $H_2O_2$ addition, HPMVEC were labeled with CellROX™ Green and imaged. C. Quantification of ROS from COVID-19 and normal serum by integrating fluorescent signals using Image J, with fluorescence intensity normalized to field area of the cells. Data are shown as a box & whiskers plot. Data were obtained from N = 4 of each category (COVID-19 affected and healthy individuals). Three independent experiments were carried out for each individual's serum sample and the average of the three trials was considered to represent the values for that sample. $^*p \leq 0.05$ when compared to healthy serum.

fluorescence upon excitation at 488 nm. Fig 5A shows representative images of ROS production observed across all samples. Untreated and $H_2O_2$-treated HPMVEC were used as the negative and positive controls, respectively. The positive control showed that the localization of the fluorescence signal depends the amount of local ROS. High $H_2O_2$ showed ROS signal in nuclear regions (Fig 5B). Fig 5C shows that HPMVEC produced significantly higher ROS levels when treated with COVID-19 serum as compared to normal serum (healthy controls).

## NOX2 activation as the source of ROS

The source of ROS in the HPMVECs with COVID-19 is unclear; therefore, we assessed TNF-α levels in all serum samples and found that it was within the range of 11.2–18.5 pg/ml for patients with COVID-19 and undetectable in healthy controls. We have previously shown that in pulmonary endothelial cells, NADPH oxidase 2 (NOX2) is the major source of ROS production in response to certain stimuli such as mechanical stress and chemical agents (like LPS, TNF-α) [29, 30]. To identify the source of endothelial ROS in COVID-19, we used PIP-2,

which blocks NOX2 assembly. HPMVECs were pre-treated with PIP-2 encapsulated in liposomes (PIP-2 carrying liposomes) before exposure to COVID-19 serum. Blank liposomes were used as controls. Representative images (Fig 6A and 6C) show ROS production in the presence of the NOX2 inhibitor PIP-2 as detected using two dyes, CellROX™ Green and DHE. The effect of PIP-2 pre-treatment on ROS was measured using DHE, a semi-permeable dye oxidized by superoxide ($O_2$-.) to oxyethidium (2-OHE); and oxidized by other ROS into ethidium (E+). 2-OHE was visualized by fluorescence at $\lambda$ex 520 nm/$\lambda$em 590–620 nm. Pre-treatment with PIP-2 significantly reduced ROS production (Fig 6B and 6D).

NOX2 activation was also monitored by measuring the translocation of the Rac subunit (Fig 7A) to the plasma membrane of HPMVEC. Rac translocation from the cytosol to the membrane enables its assembly with the other NOX2 membrane components to produce ROS. Rac translocation can thus be considered as a surrogate for NOX2 activation and ROS production. Rac was measured by monitoring the fluorescence intensity of Rac1 (green) on the EC membranes. For each cell, the cell membrane was outlined as a dotted line (cell perimeter) and the green fluorescence intensity was normalized to the cell perimeter using Image J. As shown in Fig 7B, PIP-2 significantly reduced Rac translocation.

## The role of NOX2-ROS in regulating of endothelial inflammation

We previously reported that ROS initiates endothelial signaling that drives a pro-inflammatory phenotype in ECs [10, 25, 27, 31, 32]. Therefore, we examined the effect of NOX2 blockade on adhesion molecules (ICAM-1). PIP-2 pre-treatment was observed to reduce ICAM-1 expression in HPMVEC (Fig 8).

## The role of NOX2-ROS in the regulating the NLRP3 inflammasome

Our recent inspections of the lungs of patients with COVID-19 revealed a significantly high expression of NLRP3 inflammasome along the around the vessel walls (endothelial layer) as compared to in lungs from non-COVID-19 affected individuals [15]. Thus, we assessed the regulation of the NLRP3 subunit and its downstream effector caspase 1, in the *in vitro* HPMVEC COVID-19 model. A significant increase in NLRP3 and caspase-1 expression was observed in HPMVEC treated with COVID-19 serum compared to those treated with healthy serum (Fig 9A and 9B). PIP-2 pre-treatment led to significant blunting of the NLRP3 pathway. Thus, both the NLRP3 subunit and caspase-1 appear to be regulated by NOX2-induced ROS (Fig 9C).

## The role of NOX2-ROS in endothelial cell death in the *in vitro* COVID-19 model

Cell death was measured in the presence of the NOX2 inhibitor, PIP-2. Pre-treatment with PIP-2 led to a reduction in cell death as compared to cells treated with blank liposomes (Fig 10).

## Discussion

The vascular system tightly regulates various functions through the endothelium, such as vasomotion, inflammation, oxidative stress response, vascular permeability and structure maintenance [33, 34]. The endothelial layer by virtue of being an interface between blood and tissues responds to mechanical and chemical cues of the systemic circulation by triggering signaling cascades that either maintain homeostasis or initiate cellular changes that drive diseased pathologies [29, 30]. Indeed, our earlier work provided insights into the processes by which

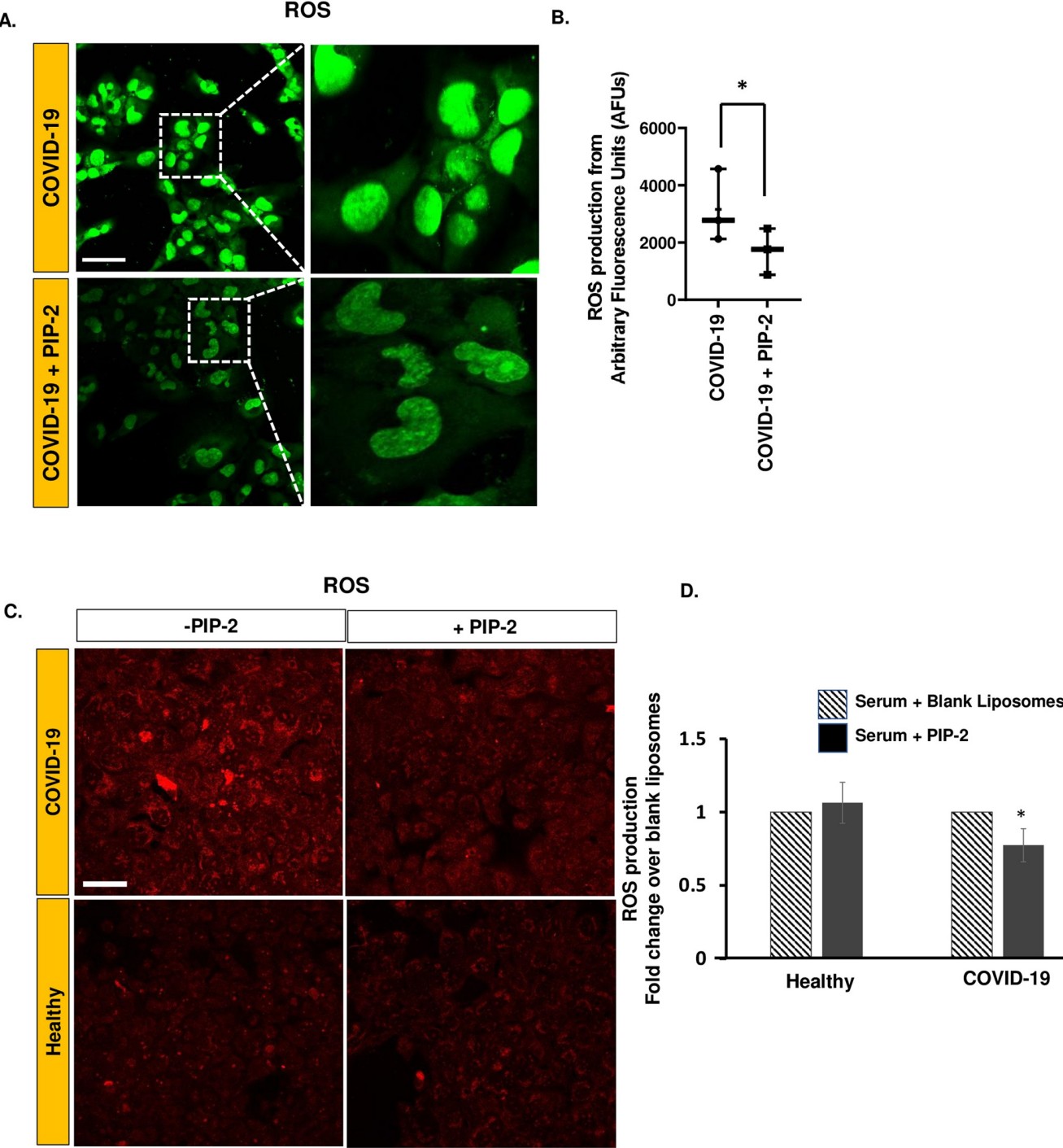

**Fig 6. Effect of PIP-2 (NOX2 inhibitor) on Reactive Oxygen Species (ROS) production.** ROS was measured in separate experiments using two different dyes CellROX™ Green and dihydroethidium (DHE). HPMVEC were pre-treated with either blank liposomes or PIP-2 carrying liposomes for 3 h followed by (media supplemented with) COVID-19 serum for 1 h. A. Cells were then labeled with CellROX™ Green and imaged for fluorescence. Scale bar 25 μm. "COVID-19" denotes cells pre-treated with blank liposomes followed by COVID-19 serum; "COVID-19 + PIP-2" denotes cells pre-treated with PIP-2 carrying liposomes followed by COVID-19 serum. B. Quantification of the Cell-ROX™ Green fluorescent signal by Image J. The integrated intensity in a field was normalized to the field area (i.e., area occupied by cells within the field). Data are shown as a box & whiskers plot. Data were obtained from n = 3 samples in each category (COVID-19 and COVID-19 + PIP-2). *denotes p≤ 0.05 when compared COVID-19. C. Superoxide by DHE: Cells were exposed for 1 h to (media supplemented with) COVID-19 serum or normal serum (from healthy individuals). In separate experiments, cells were pre-treated with either blank liposomes or PIP-2 carrying liposomes for 3 h followed by media supplemented with COVID-19 serum. This was followed by labeling with DHE (10 μM) which is oxidized by superoxide to form 2-hydroxyethidium (2-OH-E+) (λex 520 nm/λem 590–620 nm). Scale bar = 50 μm. "- PIP2" denotes

cells pre-treated with blank liposomes followed by COVID-19 serum; "+ PIP-2" denotes cells pre-treated with PIP-2 carrying liposomes followed by COVID-19 serum. D. Quantification of the DHE fluorescent signal (red). A total of n = 5 serum samples of each cohort (COVID-19/healthy or blank liposomes/PIP-2 carrying liposomes) were assessed. For each serum sample, three independent experiments were carried out. For each experiment 3–4 fields were summarized. Data were obtained as arbitrary fluorescence units and were normalized to fold increase over -PIP2 (blank liposomes followed by COVID-19 serum). Data are presented as mean ± SD of n = 5 samples each with and without PIP-2. *denotes p≤ 0.05 when compared to COVID-19.

chemical and mechanical cues emanating from the systemic circulation are "sensed" by the ECs leading to the initiation of biochemical signals associated with inflammation, oxidative stress, proliferation and angiogenesis [30, 32]. In addition, ECs have been reported to detect pathogens and to trigger the host defenses by sensing its presence and sending warning signs of infection, invasion or injury [35].

COVID-19 is associated with pervasive EC injury. In the lungs, endothelial injury during the progression from respiratory disease to acute respiratory distress syndrome (ARDS) is largely driven by extensive immune cell recruitment. However, the signaling mechanisms that cause lung endothelial activation and endothelial inflammation in COVID-19 are not clear. COVID-19 primarily affects the pulmonary system and causes lung vascular damage. Indeed, we observed extensive endothelial damage in the postmortem lung tissue of patients with

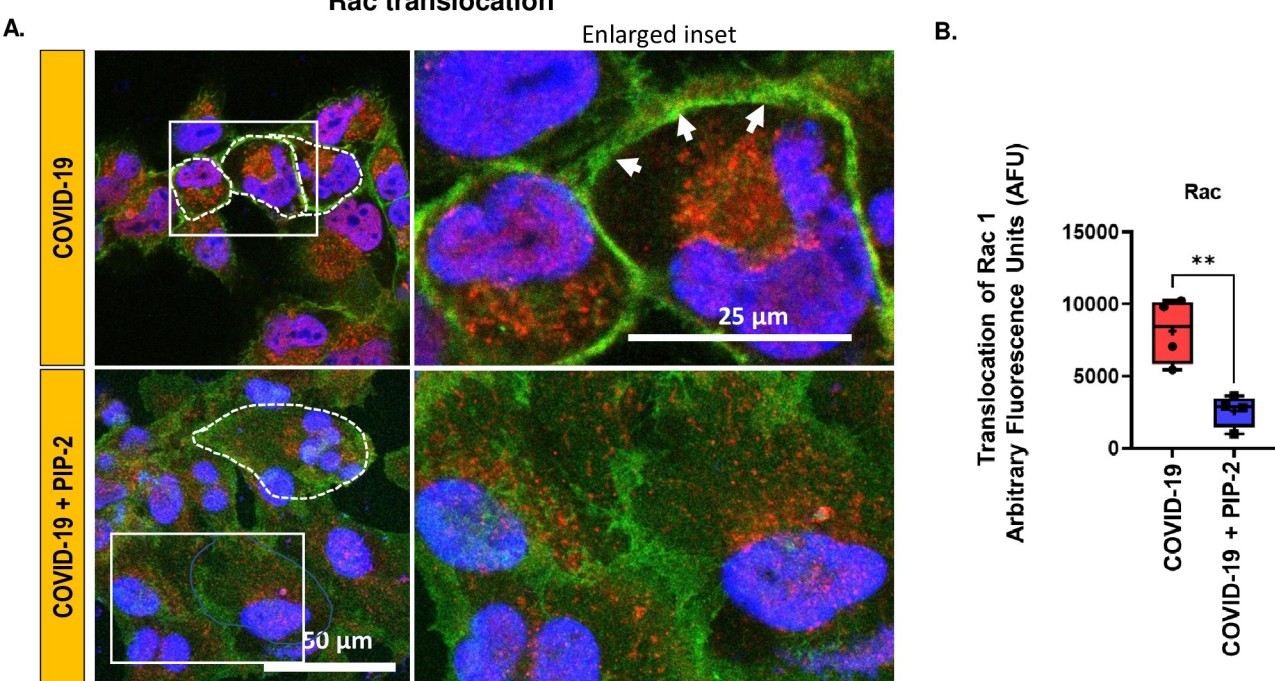

**Fig 7. Rac1 translocation to endothelial membrane.** A. Immunofluorescent labeling of Rac 1 (green) and membrane marker flotillin (red) in HPMVEC. HPMVEC were pre-treated with blank liposomes or PIP-2 carrying liposomes for 3 h followed by 1 h incubation with (media supplemented with) COVID-19 serum. Cells were fixed and immunostained with anti-Rac-GFP (1:100, green) and anti-flotillin (1:250). The secondary antibody was conjugated to Alexa 594 (1:200; red). DAPI (blue) was used to stain the nuclei. Rac1 is visualized as a green signal. Membranes were visualized using flotillin (red). The yellow colabel (white arrow) indicates the colocalization of Rac1 with the EC membrane. "COVID-19" denotes cells pre-treated with blank liposomes followed by COVID-19 serum; "COVID-19 + PIP-2" denotes cells pre-treated with PIP-2 carrying liposomes followed by COVID-19 serum. B. Rac translocation was analyzed by the ImageJ program. For each field, the perimeters of 3–4 cells were outlined (as shown by the dotted line). The green fluorescence intensity (Rac signal) along the cell border was measured and normalized to the cell perimeter. Three separate experiments were conducted for each sample. The average number from the three independent experiments was considered to represent the values for that sample. Data were obtained from N = 5 samples for each category (COVID-19 +PIP2 and COVID-19). Data are shown as a box & whiskers plot. The group average and median are indicated by a plus sign and horizontal bar, respectively. *p<0.05 as compared to COVID-19.

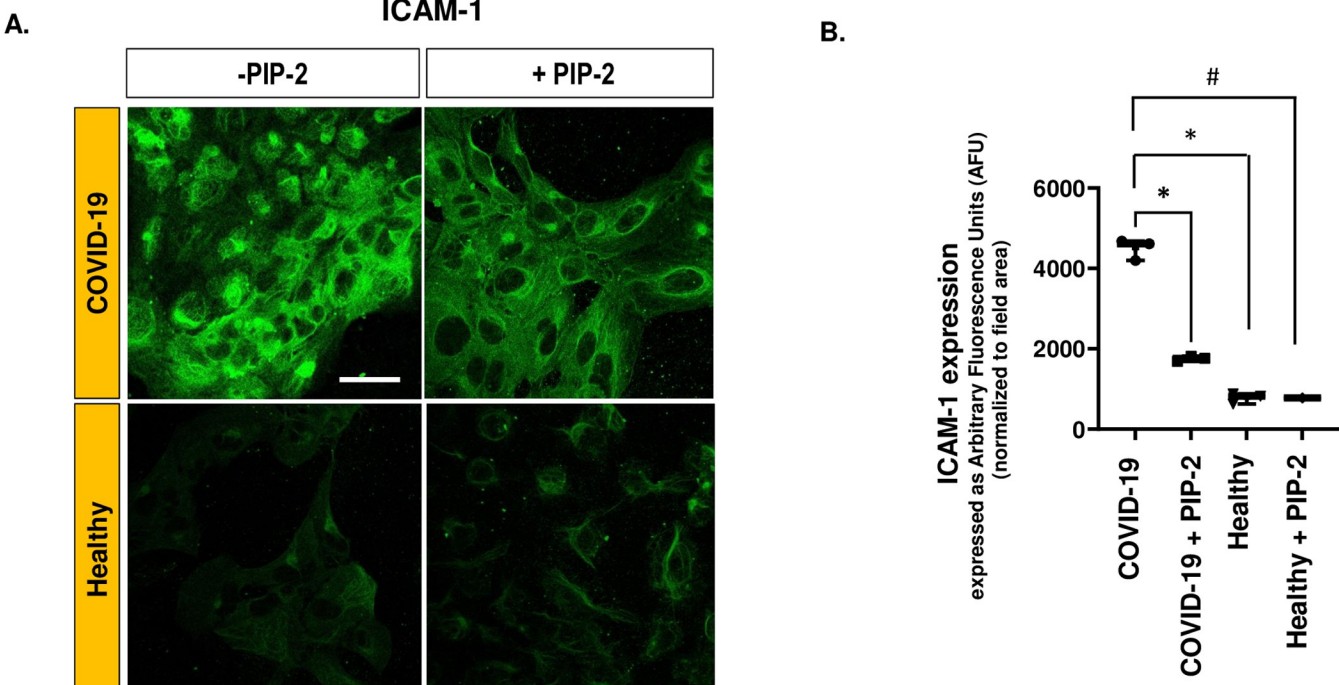

**Fig 8. PIP-2 blocks ICAM-1 expression in an in vitro model of HPMVEC COVID-19.** A. HPMVEC were treated with either blank liposomes or PIP-2 carrying liposomes for 3 h and then treated with COVID-19 serum or normal serum (from healthy individuals) for 2 hrs. Cells were then fixed, and immunostained to measure ICAM-1 expression. Scale bar = 25 μm. B. Quantification of the fluorescent signals using ImageJ. The green fluorescence intensity (representing ICAM-1 expression) was integrated across each field (area occupied by cells within the field). "-PIP-2" denotes cells pre-treated with blank liposomes followed by COVID-19 serum; "+ PIP-2" denotes cells pre-treated with PIP-2 carrying liposomes. Data are shown as a box & whiskers plot. Data were obtained from n = 3 samples in each category (COVID-19, COVID-19 + PIP-2, healthy, healthy + PIP-2). * p≤ 0.05 and #p≤ 0.001 as compared to COVID-19.

COVID-19 [15]. As a follow up to that study, our goal here was to determine the mechanisms why which COVID-19 drives EC activation or inflammation (i.e., a pro-inflammatory phenotype).

We recreated the "inflammatory load" of the systemic circulation that is faced by the lung endothelium. SARS-CoV-2 virus does not generally directly infect the endothelial cells; however, it infects epithelial cells, triggering an inflammation-immune cascade. The resultant cytokine storm in the circulation affects ECs because these cells are continuously exposed to blood flow. Therefore, COVID-19's effect on the endothelium is indirect. The *in vitro* model used in this study comprised HPMVECs in media supplemented with human serum from patients with COVID-19. Studies based on applying serum (from affected patients) to endothelial cells in culture and measuring the cellular responses have been reported for other pathologies such as obstructive sleep apnea and Alzheimer's disease [36, 37].

Our data reveal that the endothelial cells that are in contact with COVID-19 serum, are exposed to the cytokine TNF-α that is in the systemic circulation of COVID-19 patients. TNF-α has been reported to activate ECs in other vascular beds to produce ROS [38, 39]. With COVID-19, ECs produced significantly higher levels of ROS (both $H_2O_2$ and superoxide, as evidenced by increased fluorescence of ROS and superoxide sensitive dyes) as compared to healthy controls. CellROX™ Green fluorescence revealed the localization of ROS in the nuclear and in the perinuclear membranes. We have previously reported that ROS are produced by several cellular compartments in HPMVEC [16, 25]. The source of ROS in these cells was

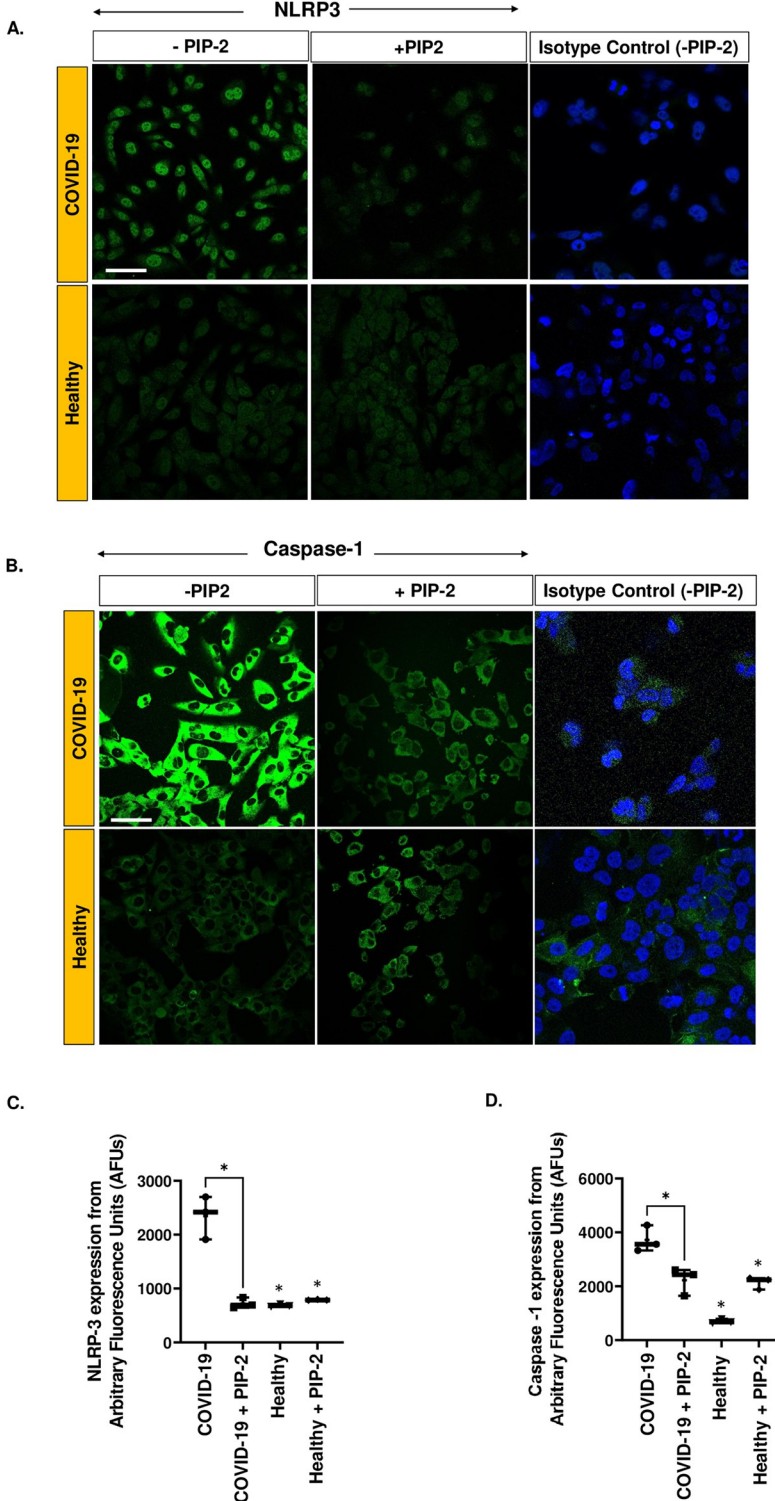

**Fig 9. The role of NOX2-ROS in regulating the NLRP3 inflammasome in COVID-19.** NLRP3 subunit and caspase-1 in HPMVEC after exposure to COVID-19 serum or normal serum (from healthy individuals), with or without PIP-2 pre-treatment. HPMVEC were treated with either blank liposomes or PIP-2 carrying liposomes for 3 h and then treated with COVID-19 serum or normal serum for 2 hrs. Cells were then fixed, and immunostained for A. NLRP3 using anti-NLRP3 B. Caspase-1 using antibody against activated caspase. Scale bar = 25 μm. C. Quantification of NLRP3 and caspase-1 was done by quantifying the intensity of the fluorescence signal (normalized to area occupied by

cells) using Image J. "COVID-19" denotes cells pre-treated with blank liposomes followed by COVID-19 serum; "COVID-19 + PIP-2" denotes cells pre-treated with PIP-2 carrying liposomes followed by COVID-19 serum; "Healthy" denotes cells pre-treated with blank liposomes followed by serum from healthy individuals; "Healthy+ PIP-2" denotes cells pre-treated with PIP-2 carrying liposomes followed by serum from healthy individuals. Data are shown as a box & whiskers plot. Data was obtained from n = 3 subjects in each category (COVID-19, COVID-19 + PIP-2, healthy, healthy + PIP-2). *p≤0.05 as compared to COVID-19.

observed to be the plasma membrane NOX2, which is activated when its cytosolic components (Rac, p67phox, p47phox etc.) assembled with the membrane components (gp91phox and p22phox) to produce superoxide outside the membrane. Superoxide either dismutates to $H_2O_2$ (which can diffuse across membranes into the cell) or enters the cell via ion channels and activates mitochondrial ROS production. From these studies, we concluded that plasma NOX2 activation has the potential to induce mitochondrial ROS production. Here we speculate that i) the inflammatory stimulus of COVID-19 induced NOX2 activation may lead to ROS production in the nuclear and perinuclear regions as well and ii) high amounts of ROS produced in the COVID-19 model can induce membrane damage that prevents ROS-sensitive dyes from being confined to specific cellular compartments only. This was supported by our observation that a similar localization pattern of ROS in the nuclear and perinuclear regions was observed with high (but not low) $H_2O_2$ treatment.

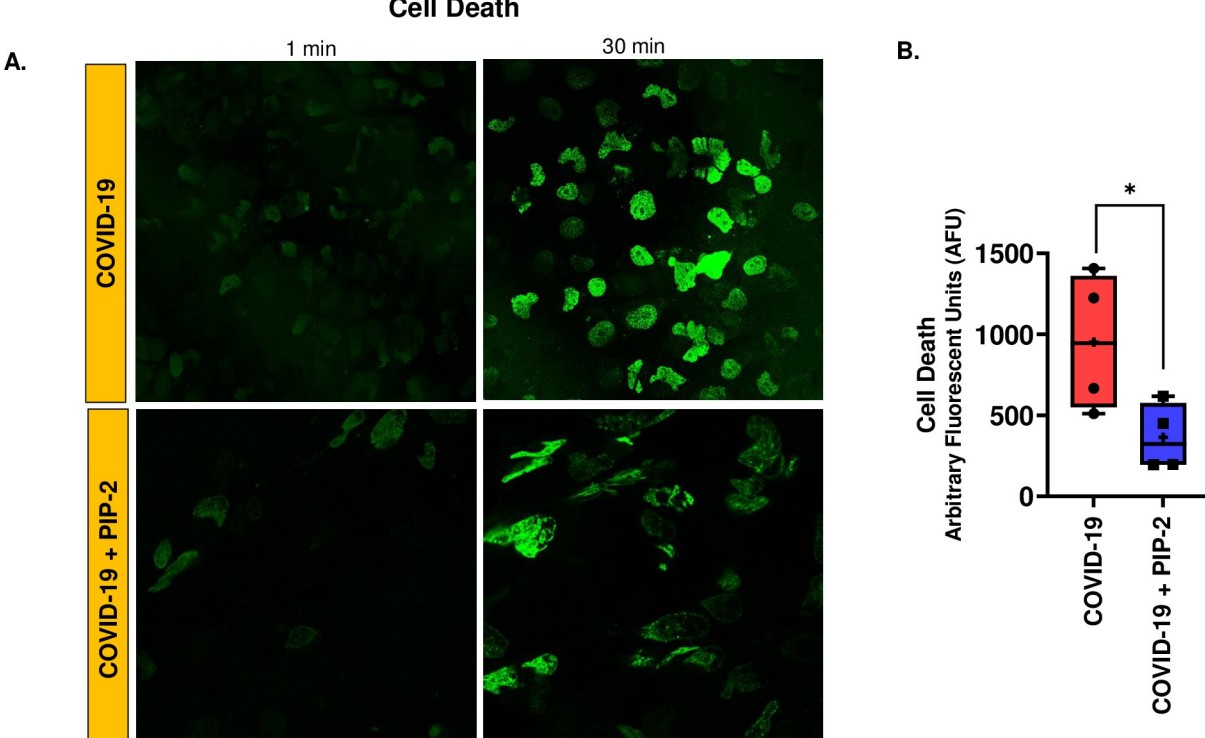

**Fig 10. NOX2-ROS in modulation of endothelial cell death in an in vitro model of COVID-19.** HPMVEC were treated with either blank liposomes or PIP-2 carrying liposomes for 3 h. Cells were labeled with SYTOX™ Green (100 nM) and exposed to media supplemented with COVID-19 serum for 30 min. A. Images of the same field were acquired after 1 and 30 min by confocal microscopy. Scale bar for all images is 25 μm. B. Quantification of the fluorescent signals was carried out by Image J and normalized to area of cells. Data are shown as a box & whiskers plot. The group average and median are indicated by a plus sign and horizontal bar, respectively. Data were obtained from N = 4 subjects for each category (COVID-19, COVID-19 + PIP-2). *p<0.05 as compared to COVID-19.

What are the effects of ROS production? ROS possibly evoked cell death (apoptosis or pyroptosis) as ECs in the COVID-19 model had significantly higher SYTOX™ Green fluorescence than healthy controls.

To evaluate the effect of COVID-19 on the pro-inflammatory phenotype, we monitored endothelial inflammation. Endothelial inflammation is largely regulated by ICAM-1, a cell surface glycoprotein expressed in the endothelium. In general, ICAM-1 is expressed at a low basal level in quiescent ECs but is up-regulated in response to inflammatory stimulus [40, 41]. A hallmark of inflammation is the adherence of polymorphonuclear neutrophils (PMN) and other immune cells (platelets, leukocytes) to the endothelium. This process is primarily driven by ICAM-1 and P-selectin. P-selectin is another adhesion molecule expressed (in the endothelium and platelets) with several pathologies including COVID-19 [42, 43]. We observed an increase in P-selectin expression in ECs in the COVID-19 model as compared to healthy controls, but this increase was not significant. P-selectin is reported to be shed from these cells into the systemic circulation where it induces procoagulant microparticle formation ([43, 44] The shedding of the P-selectin protein from cells could account for the lack of an appreciable increase in its expression within the cells. Another pivotal molecule that drives endothelial inflammation and injury is the NLRP3 inflammasome. It is a multiprotein complex comprised of three basic components: (1) a sensor such as a NOD-like receptor (NLR) (2) the adaptor protein apoptosis-associated speck-like protein containing a caspase-recruitment domain (ASC) and (3) the inflammatory cysteine pro-caspase 1. In certain naïve cells, the NLRP3 subunit exhibits low expression and is largely localized in the cytoplasm. Stimulus by ROS or other stimulants can lead to increased expression and translocation into the nuclear membrane. This has been reported elsewhere [45, 46]. The assembly of NLRP3 with its adaptor protein leads to the release of caspase 1 which then exerts its catalytic activity on the pro-inflammatory cytokines (IL-1β), triggering cell death, specifically inflammation induced cell death or pyroptosis [40, 41]. Both the NLRP3 subunit and caspase 1 (assessed using an antibody that detects both pro-caspase and activated caspase) were found to be significantly increased in patients with COVID-19. EC death has been reported in COVID-19. Indeed, investigations using postmortem tissues have revealed staining patterns consistent with cell death along the endothelial layer [47]. In addition, HPMVEC viability upon co-incubation with serum from COVID-19 patients was reported to be significantly higher than that upon co-incubation with serum from healthy individuals [48].

Earlier we reported that the NOX2 pathway was the predominant source of ROS in HPMVECs and other pulmonary microvascular ECs [10, 16]. We also showed that PIP-2 blocked NOX2 activation by inhibiting the PLA$_2$ activity of Prdx6 [21, 26]. In the *in vitro* COVID-19 model studied here, the NOX2 pathway (ROS, the translocation of Rac to the membrane) was blocked by PIP-2 (Figs 6 and 7). PIP-2 also blunted ICAM-1 and NLRP3 inflammasomes and cell death (Figs 8 and 9). We thus concluded that NOX2-ROS produced by the pulmonary endothelium presumably in response to the COVID-19 cytokine storm, is a key driver of endothelial inflammation and cell death during COVID-19. Thus, NOX2 inhibitors can serve as potential therapeutics for protection against damage to the endothelial layer during and perhaps, after COVID-19. Collectively, our data support the involvement of NOX2 in ROS production in COVID-19 and other models involving pulmonary endothelial cells.

We chose PIP-2 over scavengers and antioxidants because it works by blocking NOX2 assembly instead of scavenging ROS "after" it is produced (Fig 11). Antioxidants have not shown much promise predominantly because antioxidants scavenge ROS after its production and thus have to compete with endogenous cellular macromolecules in detoxification; besides, ROS quenching is never 100% efficient and unquenched ROS are sufficient to activate inflammation signaling.

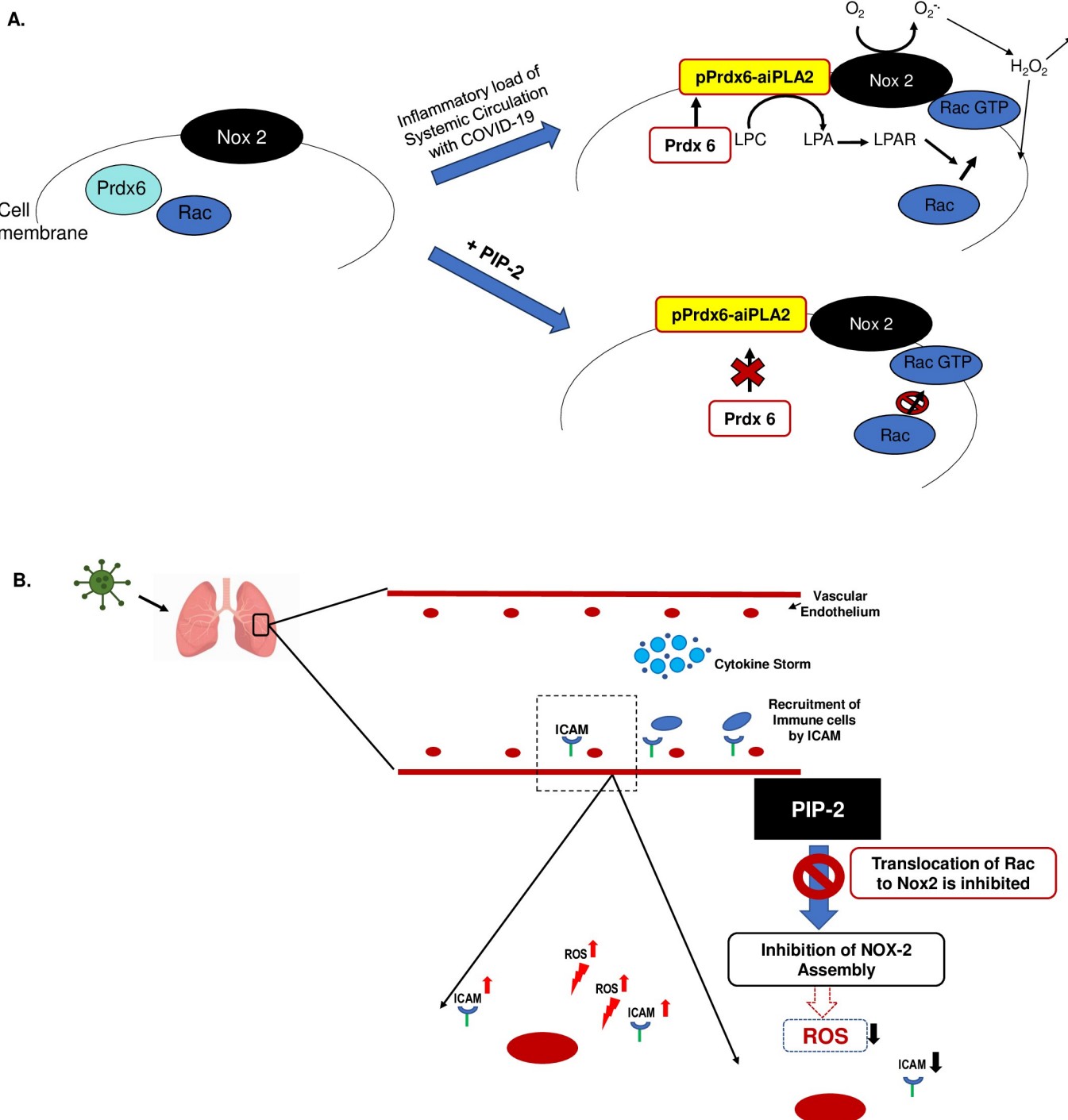

**Fig 11. Overview of endothelial inflammation in COVID-19 and the potential role of novel agent PIP-2 in endothelial protection.** A. NADPH oxidase 2 (NOX2) comprises of membrane (NOX2) and cytosolic components. Peroxiredoxin 6 (Prdx6) is known to be phosphorylated by various inflammatory stimuli (including TNF-α, which is appreciably high in COVID-19 serum). Phospho-Prdx6 (pPrdx6) translocates to the membrane and its aiPLA2 activity converts membrane phosphatidylcholine (PC) to lysophosphatidylcholine (lyso-PC). Lyso-PC is catalyzed to lysophosphatidic acid (LPA). LPA binds to its receptor on the cell membrane, and the resulting signaling cascade leads to Rac phosphorylation, which enables assembly of the cytosolic components of NOX2. The assembled enzyme reduces molecular oxygen to superoxide which then dismutates to hydrogen peroxide ($H_2O_2$) which can participate in extracellular and intracellular signaling cascades. PIP-2 inhibits the aiPLA2 activity of Prdx6 and thus blocks NOX2 activation and ROS production as reported by us earlier [26]. B. A cytokine storm is well established in the systemic circulation in COVID-19. Cytokines possibly activate the pulmonary endothelium via a NOX2 activation pathway that leads to increased expression of ICAM-1 and the NLRP3 inflammasome. ICAM-1 is known to regulate immune cell adherence while the NLRP3 pathway drives endothelial cell death (pyroptosis). Blockade of the NOX2-ROS axis by preventing NOX2 activation (that occurs via PLA₂-Rac) can abrogate this pathway potentially protecting against endothelial damage and injury.

In general, one caveat of using NOX2 blockers as therapeutics to control cellular injury *in vivo* is that such therapeutics can compromise bacterial cell killing, because PMN use NOX2 for pathogen clearance. The inhibition of ROS production can lead to a decrease in the bactericidal activity of neutrophils. Thus, in cases where neutrophils (and ROS) are the cause of lung injury that drives fatal outcomes, employing ROS blockade is a tradeoff; however, this tradeoff can be handled by using antibiotics, that is, the reduced phagocytosis can be compensated by using antibiotics. In a previous study, we reported that the increased bacterial load arising from PIP-2 administration could be reduced by antibiotics (23). This is not a problem in the case of SARS-CoV-2 as PMN play no direct role in viral clearance; indeed, in a mouse model of influenza infection, a NOX2 inhibitor, was beneficial against lung inflammation and injury [49]. However, NOX2 blockade during viral infection can drive secondary bacterial infections, in which case antibiotic administration may be necessary.

Some of the limitations of this study include its low statistical power owing to the small sample size. We attempted to overcome the caveat of small sample size by choosing to study a specific subset, that is, only those severe cases that went on to become fatal. Similarly, *in vitro* studies do not accurately represent COVID-19 *in vivo*; nevertheless, our model recreates the inflammation burden of the "cytokine storm" in the systemic circulation that affects the endothelium of an individual with COVID-19. An *in vitro* model cannot recreate the actual progression from endothelial signals to clinical manifestations of COVID-19. However, cell models can and do provide information on early endothelial signaling and hence this study was restricted to the initial and immediate endothelial responses. Indeed, our main purpose here was to identify an early signaling event in endothelial stress or damage caused by COVID-19 to be able to suggest the use of appropriate inhibitors. Currently, the non-respiratory effects of long COVID-19 are largely being attributed to endothelial dysfunction; thus, we believe that a study focusing on the blockade of aberrant vascular signaling is timely. In this context, PIP-2 was found to be efficacious as a prophylactic for the prevention of endothelial damage.

## Conclusion

Our results suggested that the systemic circulatory inflammatory load associated with COVID-19 activates the pulmonary endothelium via the NOX2-ROS pathway. This signaling pathway is an important mediator of endothelial inflammation and possibly eventual injury. Overall, this report adds to the growing list of studies on COVID-19 that highlight the importance of pulmonary vascular endothelial inflammation in fatal disease.

## Acknowledgments

We thank Nuala J. Meyer, Caroline A.G. Ittner, Ariel R. Weisman, Roseline S. Agyekum, Oluwatosin Oniyide (coordinators) and the Penn COVID processing unit for access to COVID-19 samples. The samples were collected as part of their Molecular Epidemiology of SepsiS in the ICU (MESSI-COVID) study in which patients with COVID-19 were enrolled and screened.

## Author Contributions

**Conceptualization:** Oindrila Paul, Christian Bermudez, Shampa Chatterjee.

**Data curation:** Oindrila Paul, Isha K. Akolia, Jian Qin Tao, Nikita Jain, Natalia Louneva, Shampa Chatterjee.

**Formal analysis:** Oindrila Paul, Nikita Jain, Shampa Chatterjee.

**Funding acquisition:** Shampa Chatterjee.

**Investigation:** Oindrila Paul, Kathleen T. Montone, Chamith S. Rajapakse, Shampa Chatterjee.

**Methodology:** Oindrila Paul, Isha K. Akolia, Jian Qin Tao, Shampa Chatterjee.

**Project administration:** Shampa Chatterjee.

**Resources:** Shampa Chatterjee.

**Software:** Isha K. Akolia, Shampa Chatterjee.

**Supervision:** Kathleen T. Montone, Aron B. Fisher, Chamith S. Rajapakse, Shampa Chatterjee.

**Validation:** Oindrila Paul, Jian Qin Tao, Shampa Chatterjee.

**Visualization:** Oindrila Paul, Jian Qin Tao, Shampa Chatterjee.

**Writing – original draft:** Oindrila Paul, Shampa Chatterjee.

**Writing – review & editing:** Oindrila Paul, Isha K. Akolia, Jian Qin Tao, Nikita Jain, Natalia Louneva, Kathleen T. Montone, Aron B. Fisher, Chamith S. Rajapakse, Christian Bermudez, Shampa Chatterjee.

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
