## [Decision Letter · Decision Letter 0]

21 Aug 2023

PONE-D-23-22453Reactive Oxygen Species in Endothelial Signaling with COVID-19: Role of novel peptide PIP-2 in ProtectionPLOS ONE

Dear Dr. Chatterjee,

Thank you for submitting your manuscript to PLOS ONE. After careful consideration, we feel that it has merit but does not fully meet PLOS ONE’s publication criteria as it currently stands. Therefore, we invite you to submit a revised version of the manuscript that addresses the points raised during the review process.

We look forward to receiving your revised manuscript.

Kind regards,

Eliseo A Eugenin, Ph.D.

Academic Editor

PLOS ONE

Journal Requirements:

a) The name of the colleague or the details of the professional service that edited your manuscript.

b) A copy of your manuscript showing your changes by either highlighting them or using track changes (uploaded as a *supporting information* file).

c) A clean copy of the edited manuscript (uploaded as the new *manuscript* file).

3. Please expand the acronym “NIH” (as indicated in your financial disclosure) so that it states the name of your funders in full.

"YES

SC was supported by NIH R56 HL139559 and R41HL164161. ABF is a shareholder in Peroxitech LLC, a company that is developing PIP-2 as a therapeutic agent. We thank Nuala J. Meyer (PI), Caroline A.G. Ittner, Ariel R. Weisman, Roseline S. Agyekum, Oluwatosin Oniyide (coordinators) and the Penn COVID processing unit for access to COVID-19 samples. These were part of their Molecular Epidemiology of SepsiS in the ICU (MESSI-COVID) study where COVID-19 subjects were enrolled, screened, and samples collected."

"SC was supported by NIH R56 HL139559 and R41HL164161. ABF is a shareholder in Peroxitech LLC, a

company that is developing PIP-2 as a therapeutic agent. We thank Nuala J. Meyer (PI), Caroline A.G. Ittner,

Ariel R. Weisman, Roseline S. Agyekum, Oluwatosin Oniyide (coordinators) and the Penn COVID processing

unit for access to COVID-19 samples. These were part of their Molecular Epidemiology of SepsiS in the ICU

(MESSI-COVID) study where COVID-19 subjects were enrolled, screened, and samples collected."

"YES

SC was supported by NIH R56 HL139559 and R41HL164161. ABF is a shareholder in Peroxitech LLC, a company that is developing PIP-2 as a therapeutic agent. We thank Nuala J. Meyer (PI), Caroline A.G. Ittner, Ariel R. Weisman, Roseline S. Agyekum, Oluwatosin Oniyide (coordinators) and the Penn COVID processing unit for access to COVID-19 samples. These were part of their Molecular Epidemiology of SepsiS in the ICU (MESSI-COVID) study where COVID-19 subjects were enrolled, screened, and samples collected."

6. We note that you have a patent relating to material pertinent to this article:

"I have read the journal's policy and the authors of this manuscript have the following competing interests: 

O.P., S.C. and A.B.F. have filed a U.S. Provisional Patent Application No. 63/346,663 for PIP-2 as a protective agent against endothelial damage with COVID-19."

Please provide an amended statement of Competing Interests to declare this patent (with details including name and number), along with any other relevant declarations relating to employment, consultancy, patents, products in development or modified products etc. Please confirm that this does not alter your adherence to all PLOS ONE policies on sharing data and materials, as detailed online in our guide for authors http://journals.plos.org/plosone/s/competing-interests by including the following statement: "This does not alter our adherence to  PLOS ONE policies on sharing data and materials.” 

If there are restrictions on sharing of data and/or materials, please state these. Please note that we cannot proceed with consideration of your article until this information has been declared.

7. We note that you have stated that you will provide repository information for your data at acceptance. Should your manuscript be accepted for publication, we will hold it until you provide the relevant accession numbers or DOIs necessary to access your data. If you wish to make changes to your Data Availability statement, please describe these changes in your cover letter and we will update your Data Availability statement to reflect the information you provide.

8. Your ethics statement should only appear in the Methods section of your manuscript. If your ethics statement is written in any section besides the Methods, please delete it from any other section. 

**Additional Editor Comments:**

Dear Dr. Chatterjee

thank you for submitting your manuscript to PLOSone, you manuscript was reviewer by two expert in the field and both have some suggestions and considerations. However, one reviewer and myself indicated additional concerns, listed below. Please, add all the information required and a point by point answer

1. How the staining for Fig. 1 and 2 was quantified? Negative controls with isotype are necessary

2. How were selected and catalog the blood vessels analyzed? veins, capillaries, or arteries.

3. Fig, 3, look like having a lot of autofluorescence, in the red and blue channel, as well as the green. That distribution of staining is typical of elastin. Negative controls are required.

4. NLRP3 is a protein and complex in the cytoplasm? Why most of the staining is in the nuclei. negative controls are required

5 P-selecting staining also needs controls

6. the apoptosis markers indicate that everything is going to apoptose? is this correct. How many cells are in the plate??

7 ROS Staining is wrong. ROS are mostly produced in the mitochondria or cytoplasm. Most staining is nuclear. Negative and positive controls are required

8. Fig. 7 and 8, needs controls, the pictures are overexposed. How was quantified

9. The figure legends need a better description.

Please also, check the text for a better description of the data and its interpretation

Thank you

Eliseo Eugenin

Reviewers' comments:

Reviewer's Responses to Questions

**Comments to the Author**

1. Is the manuscript technically sound, and do the data support the conclusions?

Reviewer #1: Yes

Reviewer #2: Yes

2. Has the statistical analysis been performed appropriately and rigorously? 

Reviewer #1: Yes

Reviewer #2: Yes

3. Have the authors made all data underlying the findings in their manuscript fully available?

Reviewer #1: Yes

Reviewer #2: Yes

4. Is the manuscript presented in an intelligible fashion and written in standard English?

Reviewer #1: Yes

Reviewer #2: Yes

5. Review Comments to the Author

Reviewer #1: In the introduction in the phrase "there is emerging evidence that endothelial damage occurs with COVID-19 and it impacts the outcome" is poorly worded and can be merged with the next sentence to make less"wordy".

In the results section when discussing Endothelial ICAM-1, NLRP3, P-Selectin in an in vitro model of COVID-19, you mention that P-Selectin tends to increase in COVID-19, but should probably mention that it is statistically not significant

In figure 9 Healthy serum treated cells with PIP-2 have a increase in caspase-1 that is significantly higher than the healthy, somewhat contradictory since you show that PIP-2 can decrease Caspace-1 in the COVID-19 serum treated cells.

Finally there is a type O, in the results section in the paragraph "NOX2-ROS un the regulation of the NLRP3 inflammasome, in which you refer to Covid-19 as Covid-10, and as much as I love virology i am not ready for Covid-10.....

Great work, and interesting paper.

Reviewer #2: The submitted paper by Oindrila Paul et al. gives new and interesting information about the role of endothelial cells in COVID-19 pathogenesis. Authors suggest that the cytokine storm induces the activation of NOX2 in endothelial cells via aiPLA2, inducing the expression of pro-inflammatory markers and cell death. The hypothesis was evaluated using an in vitro model (HPMVEC) exposed to serum from severe COVID-19 patients. The authors found that COVID-19 serum induced an increase in ROS, inflammatory markers, and cell death in HPMVEC cell cultures. PIP-2, a peptide that blocks NOX2 activation, prevented the COVID-19 serum effect on HPMVEC cells.

Although, in general, most results provide good data, several points need to be corrected and improved in a revised version of the manuscript.

Format

General

1. The manuscript was duplicated in the file.

2. Figures. I suggest that authors add the figures in their final format so that they can examine the context and sizes of fonts and graphics. These figures should be self-explanatory and with appropriate font sizes.

Results

1. For the three first subtitles, I suggest adding an opening line explaining the aim or why the particular analysis is being done and a concluding line for each subheading.

2. Subtitle: ICAM-1 expression along the vascular wall, Figure 2A. Check the order in the lower panel.

3. Subtitle: Endothelial ROS production in an in vitro model of COVID-19. I suggest the authors put together the results of ROS (induced by COVID-19 serum vs Healthy), CellROXTM green or Dihydroethidium (DHE) in a single figure and a second Figure showing the effect of PIP-2 on ROS production (COVID-19 vs COVID-19+PIP-2) with same format o make easy to understand for the reader.

4. About all the figures, I suggest they be ordered in the same way—for example, control vs treatment. Authors should review the format of all their graphics so that they are consistent in order, shape, and font size appropriate to the final size of the figure. In others, immunofluorescence photographs are not well aligned. Correct this.

Comments.

1. Is the effect of COVID-19 serum specific for human pulmonary microvascular endothelial cells? Or is it a general mechanism for endothelial cells?

2. Upstream, have the authors evaluated neutralizing antibodies against TNF- alpha?

3. In the manuscript, the authors show the preventive effect of PIP-2 on the inflammatory effect of CVID-19 serum. However, it could give more information about the therapeutic potential of PIP-2 if HPMVECs are simultaneously incubated with the COVID-19 serum or after its exposure to COVID-19 serum.

Minor comments

4. In the discussion section authors say, “We observed an increase in P-selectin expression in endothelial cells in the COVID-19 model as compared to healthy controls, but this increase was not significant”. The increase was not statistically significant. Therefore, there is no increase, but the authors could say they observed a tendency to increase.

This paper gives new and interesting in vitro data about the role of endothelial cells in COVID-19 pathogenesis. Most experiments are well executed, and conclusions are well presented and supported by the data. However, several results and figures described here must be improved.

6. PLOS authors have the option to publish the peer review history of their article (what does this mean?). If published, this will include your full peer review and any attached files.

Reviewer #1: **Yes: **Karl M Valerdi

Reviewer #2: No

---

## [Author Response · Author response to Decision Letter 0]

12 Feb 2024

Dear Editor and Reviewers,

We thank you for your critiques and comments. We have addressed these comments in the revised version. We regret the delay in submission. Owing to COVID-19 and long COVID-19 related health issues of lab personnel, this submission kept getting postponed. 

Below are our replies to the critiques/questions (highlighted in italics). 

Editor’s comments: 

Our response: We have edited the revised version according to the journal requirements as listed in the link. 

2. We suggest you thoroughly copyedit your manuscript for language usage, spelling, and grammar. If you do not know anyone who can help you do this, you may wish to consider employing a professional scientific editing service. Upon resubmission, please provide the following:

a) The name of the colleague or the details of the professional service that edited your manuscript.

b) A copy of your manuscript showing your changes by either highlighting them or using track changes (uploaded as a *supporting information* file).

c) A clean copy of the edited manuscript (uploaded as the new *manuscript* file).

Our response: The manuscript has been edited for language, spelling, and grammar by Elsevier Publication’s Manuscript Editing Services. 

All changes are now highlighted in red.

A clean copy has also been uploaded. 

3. Please expand the acronym “NIH” (as indicated in your financial disclosure) so that it states the name of your funders in full. This information should be included in your cover letter; we will change the online submission form on your behalf.

Our response: National Institute of Health (NIH). This information is provided in the cover letter. 

"YES

SC was supported by NIH R56 HL139559 and R41HL164161. ABF is a shareholder in Peroxitech LLC, a company that is developing PIP-2 as a therapeutic agent. We thank Nuala J. Meyer (PI), Caroline A.G. Ittner, Ariel R. Weisman, Roseline S. Agyekum, Oluwatosin Oniyide (coordinators) and the Penn COVID processing unit for access to COVID-19 samples. These were part of their Molecular Epidemiology of SepsiS in the ICU (MESSI-COVID) study where COVID-19 subjects were enrolled, screened, and samples collected."

Our response: We state in the cover letter that “The funders had no role in study design, data collection and analysis, decision to publish, or preparation of the manuscript”. 

"SC was supported by NIH R56 HL139559 and R41HL164161. ABF is a shareholder in Peroxitech LLC, a

company that is developing PIP-2 as a therapeutic agent. We thank Nuala J. Meyer (PI), Caroline A.G. Ittner,

Ariel R. Weisman, Roseline S. Agyekum, Oluwatosin Oniyide (coordinators) and the Penn COVID processing

unit for access to COVID-19 samples. These were part of their Molecular Epidemiology of SepsiS in the ICU

(MESSI-COVID) study where COVID-19 subjects were enrolled, screened, and samples collected."

"YES

SC was supported by NIH R56 HL139559 and R41HL164161. ABF is a shareholder in Peroxitech LLC, a company that is developing PIP-2 as a therapeutic agent. We thank Nuala J. Meyer (PI), Caroline A.G. Ittner, Ariel R. Weisman, Roseline S. Agyekum, Oluwatosin Oniyide (coordinators) and the Penn COVID processing unit for access to COVID-19 samples. These were part of their Molecular Epidemiology of SepsiS in the ICU (MESSI-COVID) study where COVID-19 subjects were enrolled, screened, and samples collected."

Please include your amended statements within your cover letter; we will change the online submission form SC was supported by NIH R56 HL139559 and R41HL164161. ABF is a shareholder in Peroxitech LLC, a company that is developing PIP-2 as a therapeutic agent. We thank Nuala J. Meyer (PI), Caroline A.G. Ittner, Ariel R. Weisman, Roseline S. Agyekum, Oluwatosin Oniyide (coordinators) and the Penn COVID processing unit for access to COVID-19 samples. These were part of their Molecular Epidemiology of SepsiS in the ICU (MESSI-COVID) study where COVID-19 subjects were enrolled, screened, and samples collected" on your behalf.

Our response: We have now deleted “funding related text” from the “Acknowledgement section” of the manuscript and have included an amended statement in our cover letter:

6. We note that you have a patent relating to material pertinent to this article:

"I have read the journal's policy and the authors of this manuscript have the following competing interests: 

O.P., S.C. and A.B.F. have filed a U.S. Provisional Patent Application No. 63/346,663 for PIP-2 as a protective agent against endothelial damage with COVID-19."

Our response: We have amended the statement under the subheading “Competing Interests”. It reads: 

O.P., S.C. and A.B.F. have filed a U.S. Provisional Patent Application titled “Protective Agent Against Endothelial Dysfunction”, application No. 63/346,663 filed on May 27, 2022. No related or modified products are applicable or filed. This patent application does not affect our adherence to PLOS ONE policies on sharing data, data acquisition parameters, and materials. 

Please provide an amended statement of Competing Interests to declare this patent (with details including name and number), along with any other relevant declarations relating to employment, consultancy, patents, products in development or modified products etc. Please confirm that this does not alter your adherence to all PLOS ONE policies on sharing data and materials, as detailed online in our guide for authors http://journals.plos.org/plosone/s/competing-interests by including the following statement: "This does not alter our adherence to PLOS ONE policies on sharing data and materials.” 

If there are restrictions on sharing of data and/or materials, please state these. Please note that we cannot proceed with consideration of your article until this information has been declared.

Our response: There are no restrictions on sharing data, procedures etc. and we have stated that in our “Competing Interests” section. 

7. We note that you have stated that you will provide repository information for your data at acceptance. Should your manuscript be accepted for publication, we will hold it until you provide the relevant accession numbers or DOIs necessary to access your data. If you wish to make changes to your Data Availability statement, please describe these changes in your cover letter and we will update your Data Availability statement to reflect the information you provide.

Our response: There is no repository information apart from numerous microscopy images and details on experimental methods and statistical analysis. Most of the details have been mentioned in the manuscript, although some of these details may be brief. Patient information has been de-identified. Therefore, it might be more appropriate to state that we will provide information on request from any reader. We have thus modified the data availability statement in the cover letter. 

8. Your ethics statement should only appear in the Methods section of your manuscript. If your ethics statement is written in any section besides the Methods, please delete it from any other section. 

Our response: We have now added the ethics statement to the Methods section. 

Additional Editor Comments:

Dear Dr. Chatterjee

thank you for submitting your manuscript to PLOSone, you manuscript was reviewer by two expert in the field and both have some suggestions and considerations. However, one reviewer and myself indicated additional concerns, listed below. Please, add all the information required and a point-by-point answer.

1. How the staining for Fig. 1 and 2 was quantified? Negative controls with isotype are necessary.

Our response: ICAM-1 staining was quantitated from fluorescence of green signal within the vessel area that was obtained by outlining the red signal (PECAM-1). First, the image was visualized in the red channel. Next, this area was outlined, and this outline superimposed on the green image to obtain integrated fluorescence of all the vessels. In the revised version, this is now explained in detail in the Materials and Methods section. We also describe this in the figure legend. We have added isotype controls. 

2. How were selected and catalog the blood vessels analyzed? veins, capillaries, or arteries.

Our response: All structures where PECAM-1 was positive (red fluorescence) were selected. Due to the extensive vascular damage in the lungs, identification of intact capillaries and arteries was not possible. We therefore used PECAM-1 to identify endothelium. We now state this in the Results section. 

3. Fig, 3, look like having a lot of autofluorescence, in the red and blue channel, as well as the green. That distribution of staining is typical of elastin. Negative controls are required.

Our response: We have added a panel for isotype controls. Lung tissue from COVID-19 and other subjects had large amounts of elastin so some background fluorescence was observed. But this was less at higher magnification. We mention this in the Results section. 

4. NLRP3 is a protein and complex in the cytoplasm? Why most of the staining is in the nuclei. negative controls are required. 

Our response: In certain unstimulated or naïve cells, the NLRP3 subunit shows low expression and is largely in the cytoplasm. Stimulus by ROS or other stimulants can lead to translocation into the nuclear membrane. This has been reported elsewhere (ref 1,2). 

1. Park SH, Ham S, Lee A, Möller A, Kim TS. NLRP3 negatively regulates Treg differentiation through Kpna2-mediated nuclear translocation. J Biol Chem. 2019 Nov 22;294(47):17951-17961.

2. Akbal A, Dernst A, Lovotti M, Mangan MSJ, McManus RM, Latz E. How location and cellular signaling combine to activate the NLRP3 inflammasome. Cell Mol Immunol. 2022 Nov;19(11):1201-1214. doi: 10.1038/s41423-022-00922-w. Epub 2022 Sep 20. PMID: 36127465; PMCID: PMC9622870.

Similarly, in the presence of certain agents, NLRP3 localized in the nuclear and perinuclear regions of stimulated cells, translocated to the cytoplasm. We have now expanded on NLRP3 regulation and translocation in the main text (in both Results and Discussion Section) and have added a few references. 

5. P-selecting staining also needs controls

Our response: This has been added. 

6. the apoptosis markers indicate that everything is going to apoptose? is this correct. How many cells are in the plate??

Our response: The cells are plated at a concentration of about 30-35 cell/mm2 and a large fraction of those seemed to be apoptotic by 30 min. Apoptotic cells detach and therefore for all other assays/measurements, the coverslips are lightly rinsed to remove dead/non adherent cells before measuring ROS. ICAM-1, NLRP3 or other moieties. This is now mentioned in the text (Materials and Methods Section). 

7. ROS Staining is wrong. ROS are mostly produced in the mitochondria or cytoplasm. Most staining is nuclear. Negative and positive controls are required.

Our response: We thank the Editor for this query. Based on our earlier work and reports elsewhere, ROS localization (when monitored by fluorescent dyes) cannot be accurately monitored when large amounts of ROS are produced leading to oxidative damage of membranes. 

Furthermore, we would like to place our findings in the context of our earlier work on endothelium, where ROS was observed to be produced by several cellular entities (Hawkins, 2007; Chatterjee). We showed that in the human pulmonary endothelial cells (HPMVEC), the source of ROS was plasma membrane NADPH oxidase 2 and that the activation of NADPH oxidase 2 occurred when cytosolic components (Rac, p67phox, p47phox etc.) assembled with the membrane components (gp91phox and p22phox) to produce superoxide outside the membrane. Superoxide either dismutates to H2O2 (which can diffuse across membranes into the cell), or superoxide can enter the cell via ion channels and activate mitochondrial ROS production. Thus, NADPH oxidase 2 activation has the potential to produce plasma membrane ROS that is turn drives mitochondrial ROS production. In the newly added positive (low and high H2O2) control experiments (Figure 5B), we observed a similar fluorescence in the nuclear and perinuclear regions. Therefore, we posit that with the huge inflammatory stimulus of COVID-19 may lead to i) large amounts of ROS production and associated membrane damage such that the ROS-sensitive dyes are not confined to certain cellular compartments only and ii) ROS production may also occur in nuclear and perinuclear regions as well.

The differences in ROS staining pattern are mentioned in the Results section. We include a paragraph on ROS production, ROS dye localization and staining patterns (as described in our response above) in the Discussion section.

8. Fig. 7 and 8, needs controls, the pictures are overexposed. How was quantified?

Our response: In Fig 7, Rac translocation was measured by monitoring the fluorescence intensity of Rac1 (green) on the endothelial cell membrane. For each field, 3 cells in the region of interest (ROIs) were mapped (dotted line as shown) and the fluorescent intensity of the green (Rac) signal along the dotted line normalized to the cell perimeter. This was done using Image J NIH program (in a single blinded fashion). This has been expanded and described in detail both in the Materials and Methods section and in the figure legend. 

For Fig 8, ICAM-1 expression is quantified from the fluorescence intensity. Using Image J program, integrated Intensities were normalized to the area of cells within the field as reported by us earlier (Chatterjee 2019; Chatterjee, 2021). This is now described in both the Materials and Methods section and legend to figure 8. 

Chatterjee S, Tao JQ, Johncola A, Guo W, Caporale A, Langham MC, Wehrli FW: Acute exposure to e-cigarettes causes inflammation and pulmonary endothelial oxidative stress in nonsmoking, healthy young subjects. American journal of physiology Lung cellular and molecular physiology 2019, 317:L155-L166.

Chatterjee S, Caporale A, Tao JQ, Guo W, Johncola A, Strasser AA, Leone FT, Langham MC, Wehrli FW: Acute e-cig inhalation impacts vascular health: a study in smoking naive subjects. American journal of physiology Heart and circulatory physiology 2021, 320:H144-H158.

9. The figure legends need a better description.

Please also, check the text for a better description of the data and its interpretation

Our response: We agree that the text is insufficient. In the revised version, extensive edits by us and the copy-editing service have been carried out. Specifically, we have expanded on the findings so as to allow for easier interpretation. 

Reviewers' comments:

Reviewer's Responses to Questions

Comments to the Author

1. Is the manuscript technically sound, and do the data support the conclusions?

The manuscript must describe a technically sound piece of scientific research with data that supports the conclusions. Experiments must have been conducted rigorously, with appropriate controls, replication, and

---

## [Decision Letter · Decision Letter 1]

20 Mar 2024

PONE-D-23-22453R1Reactive Oxygen Species in Endothelial Signaling in COVID-19: Protective role of the novel peptide PIP-2 in ProtectionPLOS ONE

Dear Dr. Chatterjee,

Thank you very much for submitting your manuscript to PLoS ONE. After careful consideration, we have concluded that your manuscript has the potential to be published, although some minor aspects will need to be addressed. We therefore ask that you revise your manuscript in this direction.

We look forward to receiving your revised manuscript.

Kind regards,

Gianfranco Pintus, MSc, PhD.

Academic Editor

PLOS ONE

Journal Requirements:

Reviewers' comments:

Reviewer's Responses to Questions

**Comments to the Author**

1. If the authors have adequately addressed your comments raised in a previous round of review and you feel that this manuscript is now acceptable for publication, you may indicate that here to bypass the “Comments to the Author” section, enter your conflict of interest statement in the “Confidential to Editor” section, and submit your "Accept" recommendation.

Reviewer #2: All comments have been addressed

2. Is the manuscript technically sound, and do the data support the conclusions?

Reviewer #2: Yes

3. Has the statistical analysis been performed appropriately and rigorously? 

Reviewer #2: Yes

4. Have the authors made all data underlying the findings in their manuscript fully available?

Reviewer #2: Yes

5. Is the manuscript presented in an intelligible fashion and written in standard English?

Reviewer #2: Yes

6. Review Comments to the Author

Reviewer #2: The authors have addressed all suggestions made by this reviewer.

The quality of the text improved compared to the previous manuscript.

Minor comments.

In immunofluorescence figures, some photos are not the same size or do not align well.

No further comments

7. PLOS authors have the option to publish the peer review history of their article (what does this mean?). If published, this will include your full peer review and any attached files.

Reviewer #2: No

---

## [Author Response · Author response to Decision Letter 1]

22 Mar 2024

We thank the editor, the editorial staff, and the reviewers for their support. The reviewer requested a minor change. 

Reviewer #2: 

In immunofluorescence figures, some photos are not the same size or do not align well. No further comments

Our response: We have aligned the images and have brought all images except Figure 7, where the inset is enlarged to a bigger size for visual clarity.

---

## [Editor Report · Decision Letter 2]

2 Apr 2024

Reactive Oxygen Species in Endothelial Signaling in COVID-19: Protective role of the novel peptide PIP-2 in Protection

PONE-D-23-22453R2

Dear Dr. Chatterjee,

We’re pleased to inform you that your manuscript has been judged scientifically suitable for publication and will be formally accepted for publication once it meets all outstanding technical requirements.

Kind regards,

Gianfranco Pintus, MSc, PhD.

Academic Editor

PLOS ONE
---

## [Editor Report · Acceptance letter]

4 May 2024

PONE-D-23-22453R2 

PLOS ONE

Dear Dr. Chatterjee, 

I'm pleased to inform you that your manuscript has been deemed suitable for publication in PLOS ONE. Congratulations! Your manuscript is now being handed over to our production team.

Kind regards, 

on behalf of

Dr. Gianfranco Pintus 

Academic Editor

PLOS ONE